# Analyzing the Generalization Capability of SGLD Using Properties of Gaussian Channels

**Hao Wang**
Harvard University
`hao_wang@g.harvard.edu`

**Yizhe Huang**
The University of Texas at Austin
`yizhehuang@utexas.edu`

**Rui Gao**
The University of Texas at Austin
`rui.gao@mccombs.utexas.edu`

**Flavio P. Calmon**
Harvard University
`flavio@seas.harvard.edu`

## Abstract

Optimization is a key component for training machine learning models and has a strong impact on their generalization. In this paper, we consider a particular optimization method—the stochastic gradient Langevin dynamics (SGLD) algorithm—and investigate the generalization of models trained by SGLD. We derive a new generalization bound by connecting SGLD with Gaussian channels found in information and communication theory. Our bound can be computed from the training data and incorporates the variance of gradients for quantifying a particular kind of "sharpness" of the loss landscape. We also consider a closely related algorithm with SGLD, namely differentially private SGD (DP-SGD). We prove that the generalization capability of DP-SGD can be amplified by iteration. Specifically, our bound can be sharpened by including a time-decaying factor if the DP-SGD algorithm outputs the last iterate while keeping other iterates hidden. This decay factor enables the contribution of early iterations to our bound to reduce with time and is established by strong data processing inequalities—a fundamental tool in information theory. We demonstrate our bound through numerical experiments, showing that it can predict the behavior of the true generalization gap.

## 1 Introduction

Modern deep neural networks (DNNs) are highly expressive: they can memorize an entire training dataset and still generalize well to unseen data (Zhang et al., 2016). This empirical observation is not captured by traditional generalization bounds found in statistical learning theory, which attribute the generalization ability to the use of a hypothesis class with constrained complexity (Vapnik and Chervonenkis, 1971; Valiant, 1984). Recent studies demonstrate that different algorithmic choices and data distributions may yield DNNs with contrasting generalization behaviors (Hardt et al., 2016; Neyshabur et al., 2017; Bartlett et al., 2017). In this paper, we study how one optimization method used for training DNNs, namely the stochastic gradient Langevin dynamics (SGLD) algorithm (Gelfand and Mitter, 1991; Welling and Teh, 2011), may influence their generalization.

The SGLD algorithm is used in different practical settings. For example, it has been implemented in open-source libraries (Facebook AI, 2020; Radebaugh and Erlingsson, 2019) for training models with differential privacy guarantees (Dwork et al., 2006; Song et al., 2013; Abadi et al., 2016). The additive noise in the SGLD algorithm can also mitigate overfitting for DNNs (Neelakantan et al., 2015). Recently, there is an increasing number of efforts (see e.g., Raginsky et al., 2017; Mou et al., 2018; Li et al., 2019; Pensia et al., 2018; Negrea et al., 2019) that investigate the generalization properties of the SGLD algorithm. It is within this body of work that the present paper is inscribed.

35th Conference on Neural Information Processing Systems (NeurIPS 2021).

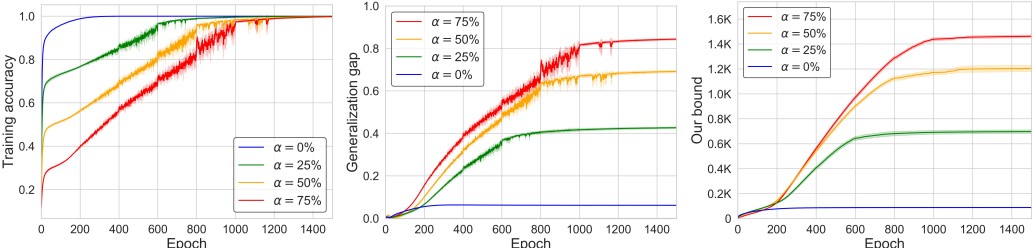

Figure 1: Illustration of our generalization bound in Theorem 1. We use the SGLD algorithm to train 3-layer neural networks on MNIST when the training data have different label corruption level $\alpha \in \{0\%, 25\%, 50\%, 75\%\}$. Left: training accuracy. Middle: (empirical) generalization gap. Right: (empirical) generalization bound. As shown, the generalization gap is increasing with respect to $\alpha$ and our bound can capture this phenomenon. We defer detailed discussions to Section 5.

We derive a new generalization bound (Theorem 1) for the SGLD algorithm in Section 3. Our bound (see Figure 1 for an illustration) incorporates the variance of gradients, which can be estimated from the training data and captures a particular kind of "sharpness" of the loss landscape (Keskar et al., 2016). This variance term can also be predictive of the generalization gap as shown in a recent empirical study (Jiang et al., 2019). We consider a general setting in which the output from the SGLD algorithm can be any function of the iterates. This is crucial since many theoretical analyses (see e.g., Zhang et al., 2017; Jin et al., 2019) require the final output to be the iterate that achieves the smallest value of the loss function or an average of the iterates, but not necessarily the last iterate. Finally, the numerical experiments in Section 5 suggest that our generalization bound is highly correlated with the true generalization gap.

We also investigate the DP-SGD algorithm in Section 4. In particular, we prove that if it is known a prior, that the algorithm outputs the last iterate rather than an arbitrary function of all iterates, then our bound can be further tightened by incorporating a time-decaying factor. Our analysis is motivated by a line of recent works (Feldman et al., 2018; Balle et al., 2019; Asoodeh et al., 2020) on *privacy amplification by iteration*. Specifically, the original work by Feldman et al. (2018) provided two intertwined observations of the DP-SGD algorithm: (i) not releasing the intermediate steps can amplify the privacy guarantees and (ii) data points used in the early iterations get stronger privacy protection than those occurring late. In this paper, we establish two analogous results: (i) our generalization bound can be sharpened by incorporating a time-decaying factor if DP-SGD only outputs the last iterate (Theorem 2) and (ii) this decay factor enables the impact of early iterations on our bound to reduce with time (Lemma 4).

The proof techniques of this paper are based on fundamental tools from information theory. We first use an information-theoretic framework, proposed by Russo and Zou (2016) and Xu and Raginsky (2017) and further tightened by Bu et al. (2020), for deriving an algorithmic generalization bound. This framework relates the generalization gap with the mutual information $I(\mathbf{W}; \mathbf{Z}_i)$ between the output parameter $\mathbf{W}$ from the SGLD algorithm and each individual data point $\mathbf{Z}_i$. However, estimating the mutual information from data is often intractable. Given this major challenge, our key contribution is to connect the SGLD algorithm with a well-understood notion in data transmission, namely additive white Gaussian noise (AWGN) channels. This connection allows us to use properties of Gaussian channels for analyzing the mutual information. First, we upper bound $I(\mathbf{W}; \mathbf{Z}_i)$ using the variance of gradients by exploring the input-output mutual information of a Gaussian channel. This variance term can be estimated from the training data and is highly correlated with the true generalization gap. Second, we incorporate a time-decaying factor into our bound. This factor is established by strong data processing inequalities (Dobrushin, 1956; Cohen et al., 1998) and has an intuitive interpretation: if a data point is used at an early iteration, its impact on the generalization gap reduces with time due to external Gaussian noise. The above two aspects correspond to Lemma 4 and Lemma 5 which, in turn, are the basis of our main results in Theorem 1 and Theorem 2.

The supplementary material of this paper includes: (i) omitted proofs of all theoretical results and (ii) supporting experimental results.

**Related Works**

We contextualize our contributions in regard to existing literature (Mou et al., 2018; Li et al., 2019; Pensia et al., 2018; Bu et al., 2020; Negrea et al., 2019; Haghifam et al., 2020; Rodríguez-Gálvez et al., 2020; Neu et al., 2021). Among them, Mou et al. (2018) introduced two generalization bounds. The first one (Theorem 1 of Mou et al., 2018), a stability-based bound, achieves $O(1/n)$ rate in terms of the sample size $n$ but relies on the Lipschitz constant of the loss function which makes it distribution-independent. A distribution-independent bound can be potentially loose and may not capture empirical observations (e.g., a network trained using true labels generalizes better than a network trained using corrupted labels as shown in Figure 1). The second one (Theorem 2 of Mou et al., 2018), a PAC-Bayes bound, replaces the Lipschitz constant by an expected-squared gradient norm but suffers from a slower rate $O(1/\sqrt{n})$. In contrast, our bound has order $O(1/n)$ and tightens the expected-squared gradient norm by the variance of gradients. The PAC-Bayes bound in Mou et al. (2018) is the only SGLD bound, besides ours, which incorporates an explicit time-decaying factor. However, their decay factor is incomparable with ours since they are established through distinct proof techniques and rely on different assumptions: their factor relies on an $L_2$ regularization[1] whereas ours is focused on the DP-SGD algorithm. A follow-up work by Li et al. (2019) combined the algorithmic stability approach with PAC-Bayesian theory and presented a bound which achieves order $O(1/n)$. However, their bound requires the scale of the learning rate to be upper bounded by the inverse Lipschitz constant of the loss function. In contrast, we do not need any assumptions on the learning rate.

There are significant recent works that adopt the information-theoretic framework (Xu and Raginsky, 2017) for bounding the SGLD generalization gap. Among them, Pensia et al. (2018) initially proposed a bound in Corollary 1 for analyzing a class of noisy iterative algorithms, including SGLD, and their bound was extended in Proposition 3 of Bu et al. (2020). However, the bounds in Pensia et al. (2018); Bu et al. (2020) are distribution-independent. Recently, Negrea et al. (2019) improved these bounds by replacing the Lipschitz constant with a gradient prediction residual, which is distribution-dependent. To compare with the generalization bound in Negrea et al. (2019), we incorporate a time-decaying factor into our bound under additional assumptions. Furthermore, the numerical experiments (Section 5) suggest that our bound is more favourably correlated with the true generalization gap. Negrea et al. (2019) provided another bound in their Theorem 3.1 which involves a variance term but it suffers from a slower rate $O(1/\sqrt{n})$.

More broadly, there are several recent works considering algorithms closely related to SGLD. For example, Haghifam et al. (2020) investigated the Langevin dynamics algorithm (i.e., full batch SGLD), which was later extended by Rodríguez-Gálvez et al. (2020) to SGLD, and observed a time-decaying phenomenon in their experiments. Specifically, Haghifam et al. (2020) incorporated a quantity, namely the squared error probability of the hypothesis test, into their bound in Theorem 4.2 and this quantity decays with the number of iterations. This seems to suggest that earlier iterations have a larger impact on their generalization bound. In contrast, our decay factor and the counterpart in Mou et al. (2018) indicate that the impact of earlier iterations is reducing with the total number of iterations. A recent work by Neu et al. (2021) investigated the generalization properties of SGD and provided a bound that also involves the variance of gradients. Note that the generalization bound in their Proposition 3 suffers from a weaker order $O(1/\sqrt{n})$ when the analysis is applied to SGLD.

To summarize, our main contribution is to provide a generalization bound for SGLD that incorporates the variance of gradients for quantifying the "sharpness" of the loss landscape and has $O(1/n)$ sample size dependence. Moreover, our bound holds under mild conditions and is applicable to a general setting in which the output from SGLD can be any function of the iterations. Under additional assumptions that are satisfied by DP-SGD, our bound includes an explicit time-decaying factor. Although each of the above contributions has been discussed in existing works, to the best of our knowledge, our bound is the first one that simultaneously combines all these aspects. Moreover, our proof techniques based on information-theoretic tools (e.g., strong data processing inequalities and properties of Gaussian channels) may be of broader interest to the community studying generalization theory.

---

[1]We note that Mou et al. (2018) also provided a bound in Theorem 23 without requiring regularization but the rate of the decay factor is slower than our factor in this case.

## 2 Preliminaries

Consider the following (possibly non-convex) optimization problem:

$$\min_{\boldsymbol{w} \in \mathcal{W}} L_\mu(\boldsymbol{w}) \triangleq \mathbb{E}\left[\ell(\boldsymbol{w}, Z)\right] = \int_{\mathcal{Z}} \ell(\boldsymbol{w}, \boldsymbol{z}) \mathrm{d}\mu(\boldsymbol{z}),$$

where $\boldsymbol{w} \in \mathcal{W} \subseteq \mathbb{R}^d$ is the parameter (e.g., weights of a neural network) to optimize; $\mu$ is the underlying data distribution that generates Z; and $\ell : \mathcal{W} \times \mathcal{Z} \to \mathbb{R}^+$ is the loss function (e.g., 0-1 loss). Since $\mu$ is unknown, $L_\mu(\boldsymbol{w})$ cannot be computed directly. Hence, one can instead minimize the empirical risk using a dataset $S \triangleq (Z_1, \cdots, Z_n)$ which contains $n$ i.i.d. points $Z_i \sim \mu$:

$$\min_{\boldsymbol{w} \in \mathcal{W}} L_S(\boldsymbol{w}) \triangleq \frac{1}{n} \sum_{i=1}^{n} \ell(\boldsymbol{w}, Z_i).$$

**Stochastic gradient Langevin dynamics.**   We consider the stochastic gradient Langevin dynamics (SGLD) algorithm (Gelfand and Mitter, 1991; Welling and Teh, 2011) for solving this empirical risk optimization. The dataset S is first divided into $m$ disjoint mini-batches:

$$S = \bigcup_{j=1}^{m} S_j, \quad \text{where } |S_j| = b \text{ and } S_j \cap S_k = \emptyset \text{ for } j \neq k.$$

We initialize the parameter with a random point $W_0 \in \mathcal{W}$ and update using the following rule:

$$W_t = W_{t-1} - \eta_t \nabla_{\boldsymbol{w}} \hat{\ell}(W_{t-1}, S_{B_t}) + \sqrt{\frac{2\eta_t}{\beta_t}} N, \tag{1}$$

where $\eta_t$ is the learning rate; $\beta_t$ is the inverse temperature; $N \sim N(0, \mathbf{I}_d)$ is a random variable drawn independently from a standard Gaussian distribution; $B_t \in [m]$ is the mini-batch index[2]; $\hat{\ell}$ is a surrogate loss (e.g., hinge loss); and

$$\nabla_{\boldsymbol{w}} \hat{\ell}(W_{t-1}, S_{B_t}) \triangleq \frac{1}{b} \sum_{Z \in S_{B_t}} \nabla_{\boldsymbol{w}} \hat{\ell}(W_{t-1}, Z). \tag{2}$$

The recursion in (1) runs for $T$ iterations and the final output is $W = f(W_1, \cdots, W_T)$ which is a function of the parameters across all iterations. For example, the output can be the parameter at the last iteration $W = W_T$, the one which achieves the smallest value of the loss function $W = \arg\min_{W_t} L_\mu(W_t)$, or an average of the parameters (i.e., Polyak averaging) $W = \frac{1}{T} \sum_t W_t$.

**Information-theoretic generalization bounds.**   The goal of this paper is to derive an upper bound for the *expected generalization gap*:

$$\mathbb{E}\left[L_\mu(W) - L_S(W)\right]. \tag{3}$$

A recent work by Xu and Raginsky (2017) provided a new method for bounding the expected generalization gap in terms of the mutual information between the input dataset S and the output parameter W. The bound in Xu and Raginsky (2017) was later tightened by Bu et al. (2020).

**Lemma 1** (Bu et al. (2020) Proposition 1). *Let the loss function $\ell(\boldsymbol{w}, Z)$ be $\sigma$-sub-Gaussian under $Z \sim \mu$ for all $\boldsymbol{w} \in \mathcal{W}$. For any learning algorithm which takes a dataset $S = (Z_1, \cdots, Z_n)$ as input and outputs W,*

$$\left|\mathbb{E}\left[L_\mu(W) - L_S(W)\right]\right| \leq \frac{1}{n} \sum_{i=1}^{n} \sqrt{2\sigma^2 I(W; Z_i)}, \tag{4}$$

*where $I(W; Z_i)$ is the mutual information between the learning algorithm's output W and an individual data point $Z_i$.*

---

[2]For the sake of illustration, we assume that the mini-batch indices are specified before the SGLD is run.

**Strong data processing inequalities.** In order to characterize the time-decaying phenomenon, we use an information-theoretic tool: strong data processing inequalities (Dobrushin, 1956; Cohen et al., 1998). The data processing inequality (Cover and Thomas, 2012) states that if a Markov chain $U \to X \to Y$ holds, then $I(U;Y) \leq I(U;X)$. In other words, no post-processing of X can increase the information about U. Under certain conditions, the data processing inequality can be sharpened, which leads to a strong data processing inequality, often cast in terms of a contraction coefficient. Next, we recall the contraction coefficients of $f$-divergences and show their connection with strong data processing inequalities.

Let $f : (0, \infty) \to \mathbb{R}$ be a convex function with $f(1) = 0$ and $P, Q$ be two probability distributions over a set $\mathcal{X} \subseteq \mathbb{R}^d$. The $f$-divergence (Csiszár, 1967) between $P$ and $Q$ is defined as

$$\mathrm{D}_f(P\|Q) \triangleq \int_{\mathcal{X}} f\left(\tfrac{\mathrm{d}P}{\mathrm{d}Q}\right) \mathrm{d}Q. \tag{5}$$

Examples of $f$-divergence include KL-divergence ($f(t) = t \log t$) and total variation distance ($f(t) = |t - 1|/2$). For a given transition probability kernel $P_{Y|X} : \mathcal{X} \to \mathcal{Y}$, let $P_{Y|X} \circ P$ be the distribution on $\mathcal{Y}$ induced by the push-forward of the distribution $P$ (i.e., the distribution of Y when the distribution of X is $P$). The contraction coefficient of $P_{Y|X}$ for $\mathrm{D}_f$ is defined as

$$\eta_f(P_{Y|X}) \triangleq \sup_{P,Q:P \neq Q} \frac{\mathrm{D}_f(P_{Y|X} \circ P \| P_{Y|X} \circ Q)}{\mathrm{D}_f(P\|Q)} \in [0, 1].$$

In particular, when the total variation distance is used, the corresponding contraction coefficient $\eta_{\mathrm{TV}}(P_{Y|X})$ is known as the Dobrushin's coefficient (Dobrushin, 1956), which upper bounds all other contraction coefficients (Cohen et al., 1998): $\eta_f(P_{Y|X}) \leq \eta_{\mathrm{TV}}(P_{Y|X})$. Furthermore, for any Markov chain $U \to X \to Y$, the contraction coefficient of KL-divergence satisfies (Ahlswede and Gács, 1976)

$$I(U;Y) \leq \eta_{\mathrm{KL}}(P_{Y|X}) \cdot I(U;X). \tag{6}$$

When $\eta_{\mathrm{KL}}(P_{Y|X}) < 1$, the strict inequality $I(U;Y) < I(U;X)$ improves the data processing inequality and, hence, is referred to as a strong data processing inequality. We refer the reader to Polyanskiy and Wu (2016) and Raginsky (2016) for a more comprehensive review on strong data processing inequalities and Calmon et al. (2017) for non-linear strong data processing inequalities in Gaussian channels.

**Gaussian channels.** We describe next a few fundamental properties of Gaussian channels. They will be used to derive a closed-form expression of the decay factor and to upper bound the mutual information by a quantity that can be estimated from data.

Consider a pair of random variables $(X, Y)$ related by $Y = X + m\mathrm{N}$ where X is lying on $\mathcal{X}$; $m > 0$ is a constant; and $\mathrm{N} \sim N(0, \mathbf{I}_d)$ follows a standard Gaussian distribution. This model can be regarded as a single use of a Gaussian channel, which has a long history in information theory and possesses many interesting properties. For example, if $\mathcal{X}$ is a compact set, the contraction coefficients have a non-trivial upper bound

$$\eta_{\mathrm{KL}}(P_{Y|X}) \leq \eta_{\mathrm{TV}}(P_{Y|X}) = 1 - 2\bar{\Phi}\left(\frac{\mathsf{diam}(\mathcal{X})}{2m}\right), \tag{7}$$

where $\mathsf{diam}(\mathcal{X}) \triangleq \sup_{\boldsymbol{x},\boldsymbol{x}' \in \mathcal{X}} \|\boldsymbol{x} - \boldsymbol{x}'\|_2$ is the diameter of $\mathcal{X}$ and $\bar{\Phi}(t) \triangleq \int_t^\infty \frac{1}{\sqrt{2\pi}} \exp(-v^2/2)\mathrm{d}v$ is the Gaussian complementary cumulative distribution function (CCDF). Another useful property is the following inequality (see Lemma 3.4.2 in Raginsky and Sason, 2012, for a proof) which upper bounds the KL-divergence of the output distributions from the Gaussian channel by the Wasserstein distance of their input distributions. It also serves as a fundamental lemma for proving Otto-Villani's HWI inequality (Otto and Villani, 2000) in the Gaussian case.

**Lemma 2.** *Let X and X$'$ be a pair of random variables which are independent of* $\mathrm{N} \sim N(0, \mathbf{I}_d)$. *Then for any $m > 0$*

$$\mathrm{D}_{\mathrm{KL}}(P_{X+m\mathrm{N}}\|P_{X'+m\mathrm{N}}) \leq \frac{1}{2m^2}\mathbb{W}_2^2(P_X, P_{X'}). \tag{8}$$

Here $\mathbb{W}_2(P_{\mathrm{X}}, P_{\mathrm{X}'})$ is the 2-Wasserstein distance equipped with the $L_2$ cost function:

$$\mathbb{W}_2^2(P_{\mathrm{X}}, P_{\mathrm{X}'}) \triangleq \inf \mathbb{E}\left[\|\mathrm{X} - \mathrm{X}'\|_2^2\right],$$

where the infimum is taken over all couplings (i.e., joint distributions) of the random variables $\mathrm{X}$ and $\mathrm{X}'$ with marginals $P_{\mathrm{X}}$ and $P_{\mathrm{X}'}$, respectively.

We recall an analogous result (Guo et al., 2005) which is also used in our proof. It gives an upper bound for the input-output mutual information of a Gaussian channel.

**Lemma 3.** *Let $\mathrm{X}$ be a random variable which is independent of $\mathrm{N} \sim N(0, \mathbf{I}_d)$. Then for any $m > 0$*

$$I(\mathrm{X} + m\mathrm{N}; \mathrm{X}) \leq \frac{1}{2m^2}\mathsf{Var}\left(\mathrm{X}\right). \tag{9}$$

## 3 Generalization Bounds for SGLD

Although Lemma 1 provides a generalization bound for any learning algorithm, estimating the mutual information from data is often difficult. In this section, we further upper bound the mutual information for the SGLD algorithm by using properties of Gaussian channels, discussed in the last section. This effort leads to a generalization bound (Theorem 1) which can be estimated from the training set. We observe in our experiments (Section 5) that Theorem 1 captures some generalization phenomena of DNNs, such as label corruption, and is highly correlated with the true generalization gap. We end this section by comparing our bound with existing SGLD generalization bounds (Corollary 1) and extending our analysis to a high-probability bound (Proposition 1).

Before diving into the analysis, we first discuss the main assumption made in this paper.

**Assumption 1.** *The loss function $\ell(\boldsymbol{w}, \mathrm{Z})$ is $\sigma$-sub-Gaussian under $\mathrm{Z} \sim \mu$ for all $\boldsymbol{w} \in \mathcal{W}$.*

We impose this assumption in order to apply Lemma 1. In particular, if the loss function is bounded between two constants $a$ and $b$, this assumption is naturally satisfied with sub-Gaussian constant $\sigma = (b - a)/2$.

Now we present the main result in this section—a generalization bound for the SGLD algorithm. Its proof relies on the chain rule of mutual information and properties of Gaussian channels (Lemma 3). As a reminder, the output from SGLD is allowed to be any function of the iterates (i.e., $\mathrm{W} = f(\mathrm{W}_1, \cdots, \mathrm{W}_T)$).

**Theorem 1.** *Under Assumption 1, the expected generalization gap of the SGLD algorithm has the following upper bound:*

$$\mathbb{E}\left[L_\mu(\mathrm{W}) - L_{\mathrm{S}}(\mathrm{W})\right] \leq \frac{\sqrt{2b}\sigma}{2n} \sum_{j=1}^m \sqrt{\sum_{t \in \mathcal{T}_j} \beta_t \eta_t \cdot \mathsf{Var}\left(\nabla_{\boldsymbol{w}}\hat{\ell}(\mathrm{W}_{t-1}, \mathrm{S}_j)\right)}, \tag{10}$$

*where the set $\mathcal{T}_j$ contains the indices of iterations in which the mini-batch $\mathrm{S}_j$ is used and*

$$\mathsf{Var}\left(\nabla_{\boldsymbol{w}}\hat{\ell}(\mathrm{W}_{t-1}, \mathrm{S}_j)\right) \triangleq \mathbb{E}\left[\|\nabla_{\boldsymbol{w}}\hat{\ell}(\mathrm{W}_{t-1}, \mathrm{S}_j) - \boldsymbol{e}\|_2^2\right]$$

*with the vector $\boldsymbol{e} \triangleq \mathbb{E}\left[\nabla_{\boldsymbol{w}}\hat{\ell}(\mathrm{W}_{t-1}, \mathrm{S}_j)\right]$.*

**Remark 1.** Our proof techniques can be extended to analyze a broad class of noisy iterative algorithms besides SGLD. For example, if the probability distribution of the additive noise $\mathrm{N}$ is log-Lipschitz (i.e., the logarithmic probability density function (pdf) is Lipschitz) as considered in Li et al. (2019), one can derive an analogous generalization bound by adapting our proof. In contrast, some generalization bounds (e.g., Mou et al., 2018) seem to heavily rely on the Gaussian noise.

The variance of gradients in (10) measures a particular kind of "sharpness" of the loss landscape. A recent work (Section 4.4 in Jiang et al., 2019) has observed empirically that this quantity is predictive of and highly correlated with the true generalization gap. Here we evidence this connection from a theoretical viewpoint by incorporating the gradient variance into the generalization bound.

Many existing SGLD generalization bounds (e.g., Mou et al., 2018; Li et al., 2019; Pensia et al., 2018; Negrea et al., 2019) are expressed as a sum of errors associated with each training iteration. In order to compare with these results, we present an analogous bound in the following corollary. This bound is obtained by combining a key lemma for proving Theorem 1 with Minkowski inequality and Jensen's inequality so it is often much weaker than Theorem 1.

**Corollary 1.** *Under Assumption 1, the expected generalization gap* (3) *of the SGLD algorithm can be upper bounded by*

$$\frac{\sqrt{2}\sigma}{2} \min \left\{ \frac{1}{n} \sum_{t=1}^{T} \sqrt{\beta_t \eta_t \cdot \mathsf{Var}\left(\nabla_{\boldsymbol{w}} \hat{\ell}(\mathrm{W}_{t-1}, \mathrm{Z}_t^{\dagger})\right)}, \sqrt{\frac{1}{bn} \sum_{t=1}^{T} \beta_t \eta_t \cdot \mathsf{Var}\left(\nabla_{\boldsymbol{w}} \hat{\ell}(\mathrm{W}_{t-1}, \mathrm{Z}_t^{\dagger})\right)} \right\},$$

*where $\mathrm{Z}_t^{\dagger}$ is any data point used in the $t$-th iteration.*

Our bound is distribution-dependent through the variance of gradients in contrast with Corollary 1 of Pensia et al. (2018), Proposition 3 of Bu et al. (2020), and Theorem 1 of Mou et al. (2018), which rely on the Lipschitz constant: $\sup_{\boldsymbol{w},\boldsymbol{z}} \|\nabla_{\boldsymbol{w}}\hat{\ell}(\boldsymbol{w},\boldsymbol{z})\|_2$. These bounds fail to explain some generalization phenomena of DNNs, such as label corruption (Zhang et al., 2016), because the Lipschitz constant takes a supremum over all possible weight matrices $\boldsymbol{w}$ and data points $\boldsymbol{z}$. In other words, this Lipschitz constant only relies on the architecture of the network instead of the weight matrices or data distribution. Hence, it is the same for a network trained from corrupted data and a network trained from true data. We remark that the Lipschitz constant used by Pensia et al. (2018); Bu et al. (2020); Mou et al. (2018) is different from the Lipschitz constant of the function corresponding to a network w.r.t. the input variable. The latter one has been used in the literature (see e.g., Bartlett et al., 2017) for deriving generalization bounds and, to some degree, can capture generalization phenomena, such as label corruption.

The order of our generalization bound in Corollary 1 is $\min\left(\frac{1}{n}\sum_{t=1}^{T}\sqrt{\beta\eta_t}, \sqrt{\frac{\beta}{bn}\sum_{t=1}^{T}\eta_t}\right)$. It is tighter than Theorem 2 of Mou et al. (2018) whose order is $\sqrt{\frac{\beta}{n}\sum_{t=1}^{T}\eta_t}$. Our bound is applicable regardless of the choice of learning rate while the bound in Li et al. (2019) requires the scale of the learning rate to be upper bounded by the reciprocal of the Lipschitz constant. Our Corollary 1 has the same order with Negrea et al. (2019) but we incorporate an additional decay factor under additional assumptions (see Theorem 2) and numerical experiments suggest that our bound is more favourably correlated with the true generalization gap (see Table 1).

Theorem 1 provides an upper bound for the *expected generalization gap* of the SGLD algorithm. In practice, a bound holding with high probability is also worth investigating since the SGLD algorithm may be run only once using the training set. Next, we leverage the monitor technique (Bassily et al., 2016; Xu and Raginsky, 2017) and derive a concentration inequality for the generalization gap of SGLD.

**Proposition 1.** *Under Assumption 1, with probability at least $1 - \delta$ over the randomness of $(\mathrm{S}, \mathrm{W})$, the generalization gap of the SGLD algorithm has the following upper bound:*

$$|L_\mu(\mathrm{W}) - L_\mathrm{S}(\mathrm{W})| \leq \frac{\sqrt{2b}\sigma}{n} \sum_{j=1}^{m} \sqrt{\sum_{t\in\mathcal{T}_j} \frac{\beta_t \eta_t}{\delta} \cdot \mathsf{Var}\left(\nabla_{\boldsymbol{w}}\hat{\ell}(\mathrm{W}_{t-1}, \mathrm{S}_j)\right) + \log\frac{2}{\delta}}.$$

## 4 Amplifying Generalization by Iteration for DP-SGD

In the last section, we derived a generalization bound for the SGLD algorithm and allowed the algorithmic output to be any function of all iterates. Here we consider a closely related algorithm—DP-SGD—and prove that the generalization bound can be sharpened by incorporating a time-decaying factor if the algorithmic output is the last iterate.

We start by recalling an implementation of the (projected) DP-SGD algorithm (see e.g., Algorithm 1 in Feldman et al., 2018). The parameter of the empirical risk is updated using the following rule:

$$\mathrm{W}_t = \mathsf{Proj}_{\mathcal{W}}\left(\mathrm{W}_{t-1} - \eta\left(g(\mathrm{W}_{t-1}, \mathrm{Z}_t) + \mathrm{N}\right)\right), \tag{11}$$

where $\mathrm{N} \sim N(0, \mathbf{I}_d)$ and function $g$ indicates a direction for updating the parameter. The parameter is projected $\mathsf{Proj}_{\mathcal{W}}(\boldsymbol{w}) \triangleq \mathrm{argmin}_{\boldsymbol{w}'\in\mathcal{W}} \|\boldsymbol{w}' - \boldsymbol{w}\|_2$ onto the domain $\mathcal{W}$ at each iteration. We assume

$$\sup_{\boldsymbol{w}\in\mathcal{W},\boldsymbol{z}\in\mathcal{Z}} \|g(\boldsymbol{w},\boldsymbol{z})\|_2 \leq K. \tag{12}$$

This assumption is crucial for guaranteeing differential privacy as it controls the sensitivity of each update. It is satisfied if $g$ is the gradient of a Lipschitz continuous function or the clipped gradient of a generic loss function.

The recursion in (11) is run for $T$ iterations and we assume that $T \leq n$ (i.e., data are drawn without replacement). The final output from the DP-SGD algorithm is the last iterate $W_T$. Again, our goal is to derive an upper bound for the expected generalization gap:

$$\mathbb{E}\left[L_\mu(W_T) - L_S(W_T)\right]. \tag{13}$$

Recall that Lemma 1 provides a generalization bound in terms of the mutual information $I(W_T; Z_t)$. Intuitively speaking, if a data point $Z_t$ is used at an early iteration, $I(W_T; Z_t)$ should be small due to the cumulative effect of the noise added in the iterations afterward. We characterize this intuition rigorously in the following lemma, which is established by strong data processing inequalities.

**Lemma 4.** *For the DP-SGD algorithm, we have*

$$I(W_T; Z_t) \leq I(W_t; Z_t) \cdot q^{T-t}. \tag{14}$$

*Here we define*

$$q \triangleq 1 - 2\bar{\Phi}\left(\frac{D + 2\eta K}{2\eta}\right) \in (0, 1) \tag{15}$$

*where $\bar{\Phi}(\cdot)$ is the Gaussian CCDF and the constant $D$ is the diameter of the parameter domain $\mathcal{W}$.*

Lemma 4 explains why our generalization bound in Theorem 2 incorporates a time-decaying factor. In particular, it implies that $I(W_T; Z_t) \to 0$ as $T \to \infty$. This time-decaying phenomenon occurs because the output from DP-SGD is $W_T$ while all the intermediate steps are not released. To further illustrate this point, let us imagine the opposite extreme scenario in which the DP-SGD algorithm outputs the parameters across all iterations: $W_1, \cdots, W_T$. The data processing inequality yields that for the data point $Z_t$ used at the $t$-th iteration,

$$I(W_1, \cdots, W_T; Z_t) \geq I(W_t; Z_t).$$

Hence, it is impossible to have $I(W_1, \cdots, W_T; Z_t) \to 0$ as $T \to \infty$ unless $I(W_t; Z_t) = 0$.

Since the underlying data distribution is unknown, so is the mutual information $I(W_t; Z_t)$ in (14), which poses a problem for computing the generalization bound. In order to obtain a computable bound like Theorem 1, we further upper bound the mutual information by employing properties of Gaussian channels (Lemma 2).

**Lemma 5.** *For the DP-SGD algorithm, we have*

$$I(W_t; Z_t) \leq 2 \cdot \mathsf{Var}\left(g(W_{t-1}, Z)\right), \tag{16}$$

*where the variance is over the randomness of $(W_{t-1}, Z) \sim P_{W_{t-1}} \otimes \mu$.*

With Lemma 4, 5 in hand, we now present the main result of this section—a generalization bound for the DP-SGD algorithm.

**Theorem 2.** *Under Assumption 1, the expected generalization gap of the DP-SGD algorithm has the following upper bound:*

$$\mathbb{E}\left[L_\mu(W_T) - L_S(W_T)\right] \leq \frac{2\sigma}{n} \sum_{t=1}^{T} \sqrt{\mathsf{Var}\left(g\left(W_{t-1}, Z\right)\right) \cdot q^{T-t}}, \tag{17}$$

*where $q$ is defined in (15).*

**Remark 2.** To qualitatively analyze the role of the decay factor, we demonstrate that as $T \to \infty$, the generalization bound in Theorem 2 converges to 0, whereas the counterpart without a decay factor may tend to a large constant. Under assumption (12), we can further upper bound the variance term by $K^2$. In this case, our bound in Theorem 2 becomes

$$\frac{C}{n} \sum_{t=1}^{T} \sqrt{q^{T-t}} = \frac{C}{n} \frac{1 - q^{T/2}}{1 - \sqrt{q}},$$

where the constant $C \triangleq 2\sigma K > 0$. By choosing $T = n$ and letting them go to infinity, the above bound converges to 0 but the counterpart without the decay factor can only tend to the constant $C$.

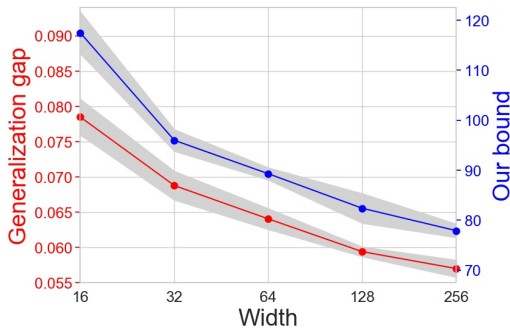

Figure 2: Comparison between the generalization gap and our generalization bound in Theorem 1. We use the SGLD algorithm to train 3-layer neural networks with varying widths on MNIST. As shown, both the generalization gap and our bound are decreasing w.r.t. the network width.

# 5 Numerical Experiments

In this section, we demonstrate our generalization bound in Theorem 1 through numerical experiments on the MNIST dataset. First, we validate the ability of our generalization bound in capturing generalization phenomena of DNNs observed by empirical studies (see e.g., Zhang et al., 2016). Second, we examine the correlation between our generalization bound and the true generalization gap using the three evaluation criteria suggested by Jiang et al. (2019), and compare it with the benchmarks. We reproduce our experiments on the CIFAR-10 dataset (Krizhevsky et al., 2009) in the supplementary material.

**Corrupted label.** As observed in Zhang et al. (2016), DNNs have the potential to memorize the entire training dataset even when a large portion of the labels are corrupted. For networks with identical architecture, those trained using true labels have better generalization capability than those ones trained using corrupted labels, although both of them achieve perfect training accuracy.

In our experiment, we randomly select 5000 samples as our training dataset and change the label of $\alpha \in \{0\%, 25\%, 50\%, 75\%\}$ training samples. Then we use the SGLD algorithm to train a 3-layer network under different corruption levels. The training process continues until the training accuracy is 1.0 (see Figure 1 Left). We compare our generalization bound with the generalization gap in Figure 1 Middle and Right. As shown, despite being vacuous, our bound is highly correlated with the true gap. When the corruption level is increasing, both our bound and the generalization gap are increasing and the curve of our bound has a very similar shape to the generalization gap. Finally, we observe that the generalization gap tends to be stable since the algorithm converges (Figure 1 Middle). Our generalization bound captures this phenomenon (Figure 1 Right) as the variance of gradients becomes negligible when the algorithm starts converging. The intuition is that the variance of gradients reflects the sharpness of the loss landscape and as the algorithm converges, the loss landscape becomes flatter.

**Network width.** As observed by several recent studies (see e.g., Neyshabur et al., 2014; Jiang et al., 2019), wider networks can lead to a smaller generalization gap. This may seem contradictory to the traditional wisdom as one may expect that a class of wider networks has a higher VC-dimension and, hence, would exhibit a higher generalize gap. In our experiments, we use the SGLD algorithm to train neural networks with different widths. The training process runs for 400 epochs until the training accuracy is 1.0. We compare our generalization bound with the generalization gap in Figure 2. As shown, both the generalization gap and our bound are decreasing with respect to the network width.

**Comparison with benchmarks.** To evaluate our bound, we adopt the three criteria proposed in Jiang et al. (2019): (i) Kendall's rank-correlation coefficient ($\tau$) (Kendall, 1938), (ii) Granulated Kendall's coefficient ($\Psi$), and (iii) conditional independent test via mutual information (MI) (Verma and Pearl, 1991). In our experiments, we select 3 commonly used hyper-parameters (i.e., learning rate (lr), width, depth), which are believed to influence the generalization gap, and let each hyper-parameter choose three different values. We train 27 neural networks under all combinations of

| dataset | method | lr | width | depth | $\tau$ | $\Psi$ | MI |
|---------|--------|-----|-------|-------|--------|--------|-----|
| MNIST | OURS (THEOREM 1) | **0.70** | **1.00** | **0.56** | **0.50** | **0.75** | **0.34** |
| | NEGREA ET AL. (2019) | 0.26 | 0.26 | 0.48 | 0.25 | 0.33 | 0.12 |

Table 1: We adopt the three evaluation criteria proposed in Jiang et al. (2019) for comparing our generalization bound with the benchmark method (Theorem 3.1 of Negrea et al., 2019): (i) Kendall's rank-correlation coefficient ($\tau$), (ii) Granulated Kendall's coefficient ($\Psi$), and (iii) conditional independent test (MI). All scores, except MI, are within $[-1, 1]$ and the score of MI is normalized to $[0, 1]$. We also report the correlations when a single hyper-parameter (e.g., learning rate (lr)) is varying.

hyper-parameters and assess the correlations between the generalization bound and the generalization gap.

We compare our generalization bound with the gradient-prediction-residual bound in Theorem 3.1 of Negrea et al. (2019) under the above three evaluation criteria. As shown in Table 1, our generalization bound is highly correlated with the true generalization gap and outperforms the benchmark under all the criteria suggested in Jiang et al. (2019).

## 6   Summary

We provide new generalization bounds for the SGLD algorithm. Our hope is that these bounds can help explain some empirical observations (e.g., why over-parameterized DNNs often generalize well in practice) and inspire new regularization methods. The proof techniques in this paper rely on information-theoretic tools (e.g., strong data processing inequalities and properties of Gaussian channels). We believe that these tools can find a wider applicability within the community studying generalization theory. Our approach can be extended in several directions. For example, one could tighten our time-decaying factor in Theorem 2 by letting it be distribution-dependent. Moreover, the proof blueprint outlined here can be applied to analyze a broader family of noisy iterative algorithms that rely on additive noise. There are several open questions that deserve further investigation. For example, we prove that our generalization bound can be tightened if the output of the algorithm is the last iterate. Our analysis is inspired by a line of works on privacy amplification by iteration (Feldman et al., 2018; Balle et al., 2019; Asoodeh et al., 2020). On the other hand, there are other ways to amplify privacy, such as subsampling (Chaudhuri and Mishra, 2006) and shuffling (Erlingsson et al., 2019). It would be interesting to understand if the algorithmic generalization capability can be improved by these methods.

## Acknowledgments and Disclosure of Funding

The work of H. Wang and F. P. Calmon is supported in part by the National Science Foundation under grants CAREER 1845852, IIS 1926925, and FAI 2040880 and F. P. Calmon also acknowledges a gift from Google Faculty Research Award and an Amazon Research Award.

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
