# A  Omitted Proofs

## A.1  Proof of Theorem 1

We first present an extension of Lemma 3 which can be proved by using the technique in Section II. E of Guo et al. (2005).

**Lemma 6.** *Let* X *be a random variable which is independent of* $N \sim N(0, \mathbf{I}_d)$. *Then for any* $m > 0$ *and function* $f$

$$I(f(X) + mN; X) \leq \frac{1}{2m^2} \mathsf{Var}\left(f(X)\right). \tag{18}$$

*More generally, if* Z *is another random variable which is independent of* N*, then for any fixed* $\boldsymbol{z}$

$$I(f(X) + mN; X \mid Z = \boldsymbol{z}) \leq \frac{1}{2m^2} \mathsf{Var}\left(f(X) \mid Z = \boldsymbol{z}\right). \tag{19}$$

*Proof.* By the property of mutual information (see Theorem 2.3 in Polyanskiy and Wu, 2019),

$$I(f(X) + mN; X) = I\left(\frac{f(X) - \boldsymbol{e}}{m} + N; X\right) \tag{20}$$

where $\boldsymbol{e} \triangleq \mathbb{E}\left[f(X)\right]$. We denote

$$g(\boldsymbol{x}) \triangleq \frac{f(\boldsymbol{x}) - \boldsymbol{e}}{m}. \tag{21}$$

The golden formula (see Theorem 3.3 in Polyanskiy and Wu, 2019, for a proof) yields

$$
\begin{aligned}
I\left(g(X) + N; X\right) &= \mathrm{D}_{\mathrm{KL}}\left(P_{g(X)+N|X}\|P_N|P_X\right) - \mathrm{D}_{\mathrm{KL}}\left(P_{g(X)+N}\|P_N\right) \\
&\leq \mathrm{D}_{\mathrm{KL}}\left(P_{g(X)+N|X}\|P_N|P_X\right).
\end{aligned} \tag{22}
$$

Furthermore, since X and N are independent, we have

$$\mathrm{D}_{\mathrm{KL}}\left(P_{g(X)+N|X=\boldsymbol{x}}\|P_N\right) = \mathrm{D}_{\mathrm{KL}}\left(P_{g(\boldsymbol{x})+N}\|P_N\right) = \frac{\|g(\boldsymbol{x})\|_2^2}{2},$$

where the last step is due to the closed-form expression of the KL-divergence between two Gaussian distributions. Finally, by the definition of conditional divergence, we have

$$\mathrm{D}_{\mathrm{KL}}\left(P_{g(X)+N|X}\|P_N|P_X\right) = \frac{1}{2}\mathbb{E}\left[\|g(X)\|_2^2\right] = \frac{1}{2m^2}\mathsf{Var}\left(f(X)\right), \tag{23}$$

where the last step is due to the definition of $g$ in (21). Combining (20–23) leads to the desired conclusion. Finally, it is straightforward to obtain (19) by conditioning on $Z = \boldsymbol{z}$ and repeating our above derivations. $\qquad \square$

Next, we present the second lemma which will be used for proving Theorem 1.

**Lemma 7.** *Under Assumption 1, the expected generalization gap* (3) *of the SGLD algorithm can be upper bounded by*

$$\frac{\sqrt{2}\sigma}{2n} \sum_{j=1}^{m} \sqrt{\sum_{t \in \mathcal{T}_j} \beta_t \eta_t \cdot \mathsf{Var}\left(\nabla_{\boldsymbol{w}}\hat{\ell}(W_{t-1}, \bar{Z}_j)\right)},$$

*where the set* $\mathcal{T}_j$ *contains the indices of iterations in which the mini-batch* $S_j$ *is used and the variance is over the randomness of* $(W_{t-1}, \bar{Z}_j) \sim P_{W_{t-1}, \bar{Z}_j}$ *with* $\bar{Z}_j$ *being any data point in the mini-batch* $S_j$.

*Proof.* We denote $Z^{(k)} \triangleq (Z_1, \cdots, Z_k)$ for $k \in [n]$ and $W^{(t)} \triangleq (W_1, \cdots, W_t)$ for $t \in [T]$. For simplicity, in what follows we only provide an upper bound for $I(W; Z_n)$. Since W is a function of $W^{(T)} = (W_1, \cdots, W_T)$, the data processing inequality yields

$$I(W; Z_n) \leq I(W^{(T)}; Z_n) \leq I(W^{(T)}, Z^{(n-1)}; Z_n). \tag{24}$$

By the chain rule,

$$I(\mathrm{W}^{(T)}, \mathrm{Z}^{(n-1)}; \mathrm{Z}_n) = I(\mathrm{W}_T; \mathrm{Z}_n \mid \mathrm{W}^{(T-1)}, \mathrm{Z}^{(n-1)}) + I(\mathrm{W}^{(T-1)}, \mathrm{Z}^{(n-1)}; \mathrm{Z}_n). \quad (25)$$

Let $\boldsymbol{w} = (\boldsymbol{w}_1, \cdots, \boldsymbol{w}_{T-1})$ and $\boldsymbol{z} = (\boldsymbol{z}_1, \cdots, \boldsymbol{z}_{n-1})$ be any two vectors. If $\mathrm{Z}_n$ is not used at the $T$-th iteration, without loss of generality we assume that the data points $\mathrm{Z}_1, \cdots, \mathrm{Z}_b$ are used in this iteration. Then

$$
\begin{aligned}
& I(\mathrm{W}_T; \mathrm{Z}_n \mid \mathrm{W}^{(T-1)} = \boldsymbol{w}, \mathrm{Z}^{(n-1)} = \boldsymbol{z}) \\
&= I\left( \boldsymbol{w}_{t-1} - \frac{\eta_T}{b} \sum_{i=1}^{b} \nabla_{\boldsymbol{w}} \hat{\ell}(\boldsymbol{w}_{T-1}, \boldsymbol{z}_i) + \sqrt{\frac{2\eta_T}{\beta_T}} \mathrm{N}; \mathrm{Z}_n \mid \mathrm{W}^{(T-1)} = \boldsymbol{w}, \mathrm{Z}^{(n-1)} = \boldsymbol{z} \right) \\
&= I\left( \mathrm{N}; \mathrm{Z}_n \mid \mathrm{W}^{(T-1)} = \boldsymbol{w}, \mathrm{Z}^{(n-1)} = \boldsymbol{z} \right) \\
&= 0. \quad (26)
\end{aligned}
$$

On the other hand, if $\mathrm{Z}_n$ is used at the $T$-th iteration, without loss of generality we assume that the other $b - 1$ data points which are also used in this iteration are $\mathrm{Z}_1, \cdots, \mathrm{Z}_{b-1}$. Then

$$
\begin{aligned}
& I(\mathrm{W}_T; \mathrm{Z}_n \mid \mathrm{W}^{(T-1)} = \boldsymbol{w}, \mathrm{Z}^{(n-1)} = \boldsymbol{z}) \\
&= I\left( \boldsymbol{w}_{t-1} - \frac{\eta_T}{b} \left( \sum_{i=1}^{b-1} \nabla_{\boldsymbol{w}} \hat{\ell}(\boldsymbol{w}_{T-1}, \boldsymbol{z}_i) + \nabla_{\boldsymbol{w}} \hat{\ell}(\boldsymbol{w}_{T-1}, \mathrm{Z}_n) \right) + \sqrt{\frac{2\eta_T}{\beta_T}} \mathrm{N}; \mathrm{Z}_n \mid \mathrm{W}^{(T-1)} = \boldsymbol{w}, \mathrm{Z}^{(n-1)} = \boldsymbol{z} \right) \\
&= I\left( -\frac{\eta_T}{b} \nabla_{\boldsymbol{w}} \hat{\ell}(\boldsymbol{w}_{T-1}, \mathrm{Z}_n) + \sqrt{\frac{2\eta_T}{\beta_T}} \mathrm{N}; \mathrm{Z}_n \mid \mathrm{W}^{(T-1)} = \boldsymbol{w}, \mathrm{Z}^{(n-1)} = \boldsymbol{z} \right). \quad (27)
\end{aligned}
$$

By Lemma 6, we have

$$
\begin{aligned}
& I\left( -\frac{\eta_T}{b} \nabla_{\boldsymbol{w}} \hat{\ell}(\boldsymbol{w}_{T-1}, \mathrm{Z}_n) + \sqrt{\frac{2\eta_T}{\beta_T}} \mathrm{N}; \mathrm{Z}_n \mid \mathrm{W}^{(T-1)} = \boldsymbol{w}, \mathrm{Z}^{(n-1)} = \boldsymbol{z} \right) \\
& \leq \frac{\beta_T \eta_T}{4b^2} \mathsf{Var}\left( \nabla_{\boldsymbol{w}} \hat{\ell}(\boldsymbol{w}_{T-1}, \mathrm{Z}_n) \mid \mathrm{W}^{(T-1)} = \boldsymbol{w}, \mathrm{Z}^{(n-1)} = \boldsymbol{z} \right). \quad (28)
\end{aligned}
$$

Substituting (28) into (27) gives

$$I(\mathrm{W}_T; \mathrm{Z}_n \mid \mathrm{W}^{(T-1)} = \boldsymbol{w}, \mathrm{Z}^{(n-1)} = \boldsymbol{z}) \leq \frac{\beta_T \eta_T}{4b^2} \mathsf{Var}\left( \nabla_{\boldsymbol{w}} \hat{\ell}(\boldsymbol{w}_{T-1}, \mathrm{Z}_n) \mid \mathrm{W}^{(T-1)} = \boldsymbol{w}, \mathrm{Z}^{(n-1)} = \boldsymbol{z} \right).$$

Taking expectation w.r.t. $(\mathrm{W}^{(T-1)}, \mathrm{Z}^{(n-1)})$ on both sides of the above inequality and using the law of total variance lead to

$$I(\mathrm{W}_T; \mathrm{Z}_n \mid \mathrm{W}^{(T-1)}, \mathrm{Z}^{(n-1)}) \leq \frac{\beta_T \eta_T}{4b^2} \mathsf{Var}\left( \nabla_{\boldsymbol{w}} \hat{\ell}(\mathrm{W}_{T-1}, \mathrm{Z}_n) \right). \quad (29)$$

To summarize, (26) and (29) can be rewritten as

$$
\begin{aligned}
& I(\mathrm{W}_T; \mathrm{Z}_n \mid \mathrm{W}^{(T-1)}, \mathrm{Z}^{(n-1)}) \\
& \leq \begin{cases} \frac{\beta_T \eta_T}{4b^2} \mathsf{Var}\left( \nabla_{\boldsymbol{w}} \hat{\ell}(\mathrm{W}_{T-1}, \mathrm{Z}_n) \right) & \text{if } \mathrm{Z}_n \text{ is used at the } T\text{-th iteration,} \\ 0 & \text{otherwise.} \end{cases} \quad (30)
\end{aligned}
$$

Assume that the data point $\mathrm{Z}_n$ belongs to the $j$-th mini-batch $\mathrm{S}_j$. Now substituting (30) into (25) and doing this procedure recursively lead to

$$I(\mathrm{W}^{(T)}, \mathrm{Z}^{(n-1)}; \mathrm{Z}_n) \leq \sum_{t \in \mathcal{T}_j} \frac{\beta_t \eta_t}{4b^2} \mathsf{Var}\left( \nabla_{\boldsymbol{w}} \hat{\ell}(\mathrm{W}_{t-1}, \mathrm{Z}_n) \right),$$

where the set $\mathcal{T}_j$ contains the indices of iterations in which the mini-batch $\mathrm{S}_j$ is used. Hence, this upper bound along with (24) naturally gives

$$I(\mathrm{W}; \mathrm{Z}_n) \leq \sum_{t \in \mathcal{T}_j} \frac{\beta_t \eta_t}{4b^2} \mathsf{Var}\left( \nabla_{\boldsymbol{w}} \hat{\ell}(\mathrm{W}_{t-1}, \mathrm{Z}_n) \right). \quad (31)$$

By symmetry, for any data point in $\mathsf{S}_j$ besides $\mathsf{Z}_n$, the mutual information between $\mathsf{W}$ and this data point can be upper bound by the right-hand side of (31) as well. Finally, recall that Lemma 1 provides an upper bound for the expected generalization gap:

$$\frac{\sqrt{2}\sigma}{n} \sum_{i=1}^{n} \sqrt{I(\mathsf{W}_T; \mathsf{Z}_i)} = \frac{\sqrt{2}\sigma}{n} \sum_{j=1}^{m} \sum_{\mathsf{Z} \in \mathsf{S}_j} \sqrt{I(\mathsf{W}_T; \mathsf{Z})}. \tag{32}$$

By substituting (31) into the above expression, we know the expected generalization gap can be further upper bounded by

$$\frac{\sqrt{2}\sigma}{2n} \sum_{j=1}^{m} \sqrt{\sum_{t \in \mathcal{T}_j} \beta_t \eta_t \cdot \mathsf{Var}\left(\nabla_{\boldsymbol{w}} \hat{\ell}(\mathsf{W}_{t-1}, \bar{\mathsf{Z}}_j)\right)},$$

where $\bar{\mathsf{Z}}_j$ is any data point in the mini-batch $\mathsf{S}_j$. $\qquad \square$

Now we are in a position to prove Theorem 1.

*Proof.* Consider a new loss function and the gradient of a new surrogate loss:

$$\ell(\boldsymbol{w}, \mathsf{S}_j) \triangleq \frac{1}{b} \sum_{\mathsf{Z} \in \mathsf{S}_j} \ell(\boldsymbol{w}, \mathsf{Z}), \quad \nabla_{\boldsymbol{w}} \hat{\ell}(\boldsymbol{w}, \mathsf{S}_j) \triangleq \frac{1}{b} \sum_{\mathsf{Z} \in \mathsf{S}_j} \nabla_{\boldsymbol{w}} \hat{\ell}(\boldsymbol{w}, \mathsf{Z}).$$

When Assumption 1 holds, $\ell(\boldsymbol{w}, \mathsf{S}_j)$ is $\sigma/\sqrt{b}$-sub-Gaussian under $\mathsf{S}_j \sim \mu^{\otimes b}$ for all $\boldsymbol{w} \in \mathcal{W}$. We view each mini-batch $\mathsf{S}_j$ as a data point and view $\ell(\boldsymbol{w}, \mathsf{S}_j)$ as a new loss function. By using Lemma 7, we obtain:

$$|\mathbb{E}\left[L_\mu(\mathsf{W}) - L_\mathsf{S}(\mathsf{W})\right]| \le \frac{\sqrt{2}\sigma}{2m\sqrt{b}} \sum_{j=1}^{m} \sqrt{\sum_{t \in \mathcal{T}_j} \beta_t \eta_t \cdot \mathsf{Var}\left(\nabla_{\boldsymbol{w}} \hat{\ell}(\mathsf{W}_{t-1}, \mathsf{S}_j)\right)}. \tag{33}$$

Since the dataset contains $n$ data points and is divided into $m$ disjoint mini-batches with size $b$, we have $n = mb$. Substituting this into (33) leads to the desired conclusion. $\qquad \square$

## A.2   Proof of Corollary 1

*Proof.* The Minkowski inequality implies that for any non-negative $x_i$, the inequality $\sqrt{\sum_i x_i} \le \sum_i \sqrt{x_i}$ holds. Therefore, we can further upper bound the generalization bound in Lemma 7 by

$$\frac{\sqrt{2}\sigma}{2n} \sum_{j=1}^{m} \sum_{t \in \mathcal{T}_j} \sqrt{\beta_t \eta_t \cdot \mathsf{Var}\left(\nabla_{\boldsymbol{w}} \hat{\ell}(\mathsf{W}_{t-1}, \bar{\mathsf{Z}}_j)\right)} = \frac{\sqrt{2}\sigma}{2n} \sum_{t=1}^{T} \sqrt{\beta_t \eta_t \cdot \mathsf{Var}\left(\nabla_{\boldsymbol{w}} \hat{\ell}(\mathsf{W}_{t-1}, \mathsf{Z}_t^\dagger)\right)}.$$

Alternatively, by Jensen's inequality and the equality $n = mb$, we can further upper bound the generalization bound in Lemma 7 by

$$\frac{\sqrt{2}\sigma}{2} \sqrt{\frac{1}{bn} \sum_{t=1}^{T} \beta_t \eta_t \cdot \mathsf{Var}\left(\nabla_{\boldsymbol{w}} \hat{\ell}(\mathsf{W}_{t-1}, \mathsf{Z}_t^\dagger)\right)}.$$

$\qquad \square$

## A.3   Proof of Lemma 4

*Proof.* For the $t$-th iteration, we can rewrite the recursion in (11) as

$$\mathsf{U}_t = \mathsf{W}_{t-1} - \eta \cdot g(\mathsf{W}_{t-1}, \mathsf{Z}_t) \tag{34a}$$
$$\mathsf{V}_t = \mathsf{U}_t + \eta \cdot \mathsf{N} \tag{34b}$$
$$\mathsf{W}_t = \mathsf{Proj}_{\mathcal{W}}(\mathsf{V}_t). \tag{34c}$$

Since data are drawn without replacement, the following Markov chain holds:

$$\mathsf{Z}_t \to \mathsf{U}_t \to \mathsf{V}_t \to \mathsf{W}_t \to \cdots \to \mathsf{W}_{T-1} \to \mathsf{U}_T \to \mathsf{V}_T \to \mathsf{W}_T.$$

Let $\mathcal{U}_T$ be the range of $\mathrm{U}_T$. By the triangle inequality,

$$\mathsf{diam}(\mathcal{U}_T) \leq \mathsf{diam}(\mathcal{W}) + 2\eta K = D + 2\eta K.$$

Now we leverage strong data processing inequalities and obtain

$$
\begin{aligned}
I(\mathrm{W}_T; \mathrm{Z}_t) &\leq I(\mathrm{V}_T; \mathrm{Z}_t) \\
&\leq \left( 1 - 2\bar{\Phi} \left( \frac{D + 2\eta K}{2\eta} \right) \right) \cdot I(\mathrm{U}_T; \mathrm{Z}_t) \\
&\leq \left( 1 - 2\bar{\Phi} \left( \frac{D + 2\eta K}{2\eta} \right) \right) \cdot I(\mathrm{W}_{T-1}; \mathrm{Z}_t),
\end{aligned}
$$

where the first and last steps are due to the data processing inequality. Applying this procedure recursively leads to the desired conclusion. $\qquad\square$

## A.4 Proof of Lemma 5

*Proof.* Recall the definition of $\mathrm{U}_t$, $\mathrm{V}_t$ in (34). The data processing inequality yields

$$I(\mathrm{W}_t; \mathrm{Z}_t) \leq I(\mathrm{V}_t; \mathrm{Z}_t). \tag{35}$$

By the definition of mutual information, we can write

$$
\begin{aligned}
I(\mathrm{V}_t; \mathrm{Z}_t) &= \mathbb{E}\left[ \mathrm{D}_{\mathrm{KL}}(P_{\mathrm{V}_t | \mathrm{Z}_t} \| P_{\mathrm{V}_t}) \right] \\
&= \int_{\mathcal{Z}} \mathrm{D}_{\mathrm{KL}}(P_{\mathrm{V}_t | \mathrm{Z}_t = \boldsymbol{z}} \| P_{\mathrm{V}_t}) \mathrm{d}\mu(\boldsymbol{z}).
\end{aligned}
\tag{36}
$$

Since $\mathrm{V}_t = \mathrm{U}_t + \eta \cdot \mathrm{N}$, Lemma 2 implies

$$\mathrm{D}_{\mathrm{KL}}\left( P_{\mathrm{V}_t | \mathrm{Z}_t = \boldsymbol{z}} \| P_{\mathrm{V}_t} \right) \leq \frac{1}{2\eta^2} \mathbb{W}_2^2 \left( P_{\mathrm{U}_t | \mathrm{Z}_t = \boldsymbol{z}}, P_{\mathrm{U}_t} \right). \tag{37}$$

To further upper bound the above Wasserstein distance, we construct a special coupling. Let $\mathrm{W}_{t-1}$ be the parameter at the $(t-1)$-st iteration. Then we introduce two random variables:

$$
\begin{aligned}
\mathrm{U}_{\boldsymbol{z}}^* &\triangleq \mathrm{W}_{t-1} - \eta \cdot g(\mathrm{W}_{t-1}, \boldsymbol{z}), \\
\mathrm{U}^* &\triangleq \mathrm{W}_{t-1} - \eta \cdot g(\mathrm{W}_{t-1}, \mathrm{Z}_t).
\end{aligned}
$$

Here $\mathrm{U}_{\boldsymbol{z}}^*$ and $\mathrm{U}^*$ have marginals, $P_{\mathrm{U}_t | \mathrm{Z}_t = \boldsymbol{z}}$ and $P_{\mathrm{U}_t}$, respectively. By the definition of Wasserstein distance, we have

$$
\begin{aligned}
\mathbb{W}_2^2 \left( P_{\mathrm{U}_t | \mathrm{Z}_t = \boldsymbol{z}}, P_{\mathrm{U}_t} \right) &\leq \mathbb{E}\left[ \| \mathrm{U}_{\boldsymbol{z}}^* - \mathrm{U}^* \|_2^2 \right] \\
&= \eta^2 \cdot \mathbb{E}\left[ \|(g(\mathrm{W}_{t-1}, \boldsymbol{z}) - g(\mathrm{W}_{t-1}, \mathrm{Z}_t))\|_2^2 \right].
\end{aligned}
\tag{38}
$$

Since $\mathrm{Z}_t$ is only used at the $t$-th iteration, it is independent of $\mathrm{W}_{t-1}$. We introduce two independent copies $\mathrm{Z}, \bar{\mathrm{Z}}$ of $\mathrm{Z}_t$ such that $(\mathrm{W}_{t-1}, \mathrm{Z}, \bar{\mathrm{Z}}) \sim P_{\mathrm{W}_{t-1}} \otimes \mu \otimes \mu$. Combining (36–38) and using Tonelli's theorem lead to

$$I(\mathrm{V}_t; \mathrm{Z}_t) \leq \frac{1}{2} \cdot \mathbb{E}\left[ \|(g(\mathrm{W}_{t-1}, \mathrm{Z}) - g(\mathrm{W}_{t-1}, \bar{\mathrm{Z}}))\|_2^2 \right]. \tag{39}$$

Now we introduce a constant vector $\boldsymbol{e} \in \mathbb{R}^d$ whose value will be specified later. Since $\mathrm{Z}, \bar{\mathrm{Z}}$ follow the same distribution and $\|\boldsymbol{a} - \boldsymbol{b}\|_2^2 \leq 2(\|\boldsymbol{a}\|_2^2 + \|\boldsymbol{b}\|_2^2)$, the right-hand side of (39) can be upper bounded by

$$2 \cdot \mathbb{E}\left[ \|g(\mathrm{W}_{t-1}, \mathrm{Z}) - \boldsymbol{e}\|_2^2 \right]. \tag{40}$$

By choosing $\boldsymbol{e} = \mathbb{E}\left[ g(\mathrm{W}_{t-1}, \mathrm{Z}) \right]$, the above quantity becomes

$$2 \cdot \mathsf{Var}\left( g(\mathrm{W}_{t-1}, \mathrm{Z}) \right). \tag{41}$$

Finally, combining (35) with (39–41) leads to the desired conclusion. $\qquad\square$

### A.5 Proof of Theorem 2

*Proof.* Lemma 1 implies that the expected generalization gap can be upper bounded by

$$\frac{1}{n}\sum_{i=1}^{n}\sqrt{2\sigma^2 I(\mathbf{W}_T;\mathbf{Z}_i)} = \frac{1}{n}\sum_{t=1}^{T}\sqrt{2\sigma^2 I(\mathbf{W}_T;\mathbf{Z}_t)}. \tag{42}$$

Lemma 4 and 5 altogether yield

$$I(\mathbf{W}_T;\mathbf{Z}_t) \leq I(\mathbf{W}_t;\mathbf{Z}_t) \cdot q^{T-t} \leq 2 \cdot \mathsf{Var}\left(g(\mathbf{W}_{t-1},\mathbf{Z})\right) \cdot q^{T-t}.$$

Consequently, the expected generalization gap can be upper bounded by

$$\frac{2\sigma}{n}\sum_{t=1}^{T}\sqrt{\mathsf{Var}\left(g(\mathbf{W}_{t-1},\mathbf{Z})\right) \cdot q^{T-t}}.$$

$\square$

### A.6 Proof of Proposition 1

The proof of Proposition 1 relies on the following lemma which can be established by a slight tweak of the proof of Theorem 3 in Xu and Raginsky (2017). We reproduce it for the sake of completeness.

**Lemma 8.** *Under Assumption 1, with probability at least $1 - \delta$, we have*

$$|L_\mu(\mathbf{W}) - L_\mathbf{S}(\mathbf{W})| \leq \frac{2\sqrt{2}\sigma}{n}\sum_{i=1}^{n}\sqrt{\frac{I(\mathbf{W};\mathbf{Z}_i)}{\delta} + \log\frac{2}{\delta}}, \tag{43}$$

*where the probability is over* $(\mathbf{S},\mathbf{W})$.

*Proof.* Let $\mathbf{S}_1, \cdots, \mathbf{S}_k$ be $k$ independent copies of the dataset $\mathbf{S}$ such that each copy $\mathbf{S}_j$ contains $n$ i.i.d. points $\mathbf{S}_j = (\mathbf{Z}_{1,j}, \cdots, \mathbf{Z}_{n,j})$. The learning algorithm is applied parallelly to each dataset $\mathbf{S}_j$ and outputs $\mathbf{W}_j$. In other words, the pairs $(\mathbf{S}_j, \mathbf{W}_j)$ with $j \in [k]$ are independent copies of $(\mathbf{S}, \mathbf{W})$. Imagine that there is a monitor which has access to the underlying distribution, the $k$ independent copies of the dataset, and the outputs from $k$ parallel algorithms. The monitor evaluates these output parameters and finds the one which overfits the most. Specifically, the monitor returns a tuple $(\mathbf{W}^*, \mathbf{J}, \mathbf{R})$ defined by

$$(\mathbf{J}, \mathbf{R}) \triangleq \operatorname*{argmax}_{j\in[k],\ r\in\{\pm 1\}} r\left(L_\mu(\mathbf{W}_j) - L_{\mathbf{S}_j}(\mathbf{W}_j)\right) \quad \text{and} \quad \mathbf{W}^* \triangleq \mathbf{W}_\mathbf{J}.$$

We view the combination of the $k$ parallel algorithms and the monitor as a new learning algorithm. This learning algorithm receives a dataset $\mathbf{S}^k \triangleq (\mathbf{Z}_1^k, \cdots, \mathbf{Z}_n^k)$ which contains $n$ i.i.d. data points $\mathbf{Z}_i^k \triangleq (\mathbf{Z}_{i,1}, \cdots, \mathbf{Z}_{i,k})$ and outputs $(\mathbf{W}^*, \mathbf{J}, \mathbf{R})$. The loss function of this new learning algorithm is $\ell : \mathcal{W} \times [k] \times \{\pm 1\} \times \mathcal{Z}^k$ defined as

$$\ell(w, j, r; \boldsymbol{z}^k) \triangleq r\ell(w, \boldsymbol{z}_j),$$

where $\boldsymbol{z}_j$ is the $j$-th coordinate of $\boldsymbol{z}^k$. When Assumption 1 holds, $\ell(w, j, r; \mathbf{Z}^k)$ is also $\sigma$-sub-Gaussian under $\mathbf{Z}^k \sim \mu^{\otimes k}$ for all $(w, j, r) \in \mathcal{W} \times [k] \times \{\pm 1\}$. Hence, applying Lemma 1 to this new learning algorithm gives

$$\mathbb{E}\left[\mathbf{R}\left(L_\mu(\mathbf{W}^*) - L_{\mathbf{S}_\mathbf{J}}(\mathbf{W}^*)\right)\right] \leq \frac{1}{n}\sum_{i=1}^{n}\sqrt{2\sigma^2 I(\mathbf{W}^*, \mathbf{J}, \mathbf{R}; \mathbf{Z}_i^k)}. \tag{44}$$

By the definition of $(\mathbf{W}^*, \mathbf{J}, \mathbf{R})$, we have

$$\mathbb{E}\left[\mathbf{R}\left(L_\mu(\mathbf{W}^*) - L_{\mathbf{S}_\mathbf{J}}(\mathbf{W}^*)\right)\right] = \mathbb{E}\left[\max_{j\in[k]}\left|L_\mu(\mathbf{W}_j) - L_{\mathbf{S}_j}(\mathbf{W}_j)\right|\right]. \tag{45}$$

Let $\mathbf{W}^k \triangleq (\mathbf{W}_1, \cdots, \mathbf{W}_k)$ be the collection of outputs from $k$ parallel algorithms. Using the chain rule for mutual information gives

$$
\begin{aligned}
I(\mathbf{W}^*, \mathbf{J}, \mathbf{R}; \mathbf{Z}_i^k) &\leq I(\mathbf{W}^k, \mathbf{W}^*, \mathbf{J}, \mathbf{R}; \mathbf{Z}_i^k) \\
&= I(\mathbf{W}^k; \mathbf{Z}_i^k) + I(\mathbf{W}^*, \mathbf{J}, \mathbf{R}; \mathbf{Z}_i^k \mid \mathbf{W}^k) \\
&= \sum_{j=1}^{k} I(\mathbf{W}_j; \mathbf{Z}_{i,j}) + I(\mathbf{W}^*, \mathbf{J}, \mathbf{R}; \mathbf{Z}_i^k \mid \mathbf{W}^k),
\end{aligned}
$$

where the last step is because of the independence among $(\mathbf{W}_j, \mathbf{Z}_{i,j})$ for $j \in [k]$. Since $(\mathbf{W}^*, \mathbf{J}, \mathbf{R})$ can take at most $2k$ different values given $\mathbf{W}^k$, then

$$
I(\mathbf{W}^*, \mathbf{J}, \mathbf{R}; \mathbf{Z}_i^k \mid \mathbf{W}^k) \leq \log(2k).
$$

By symmetry, $I(\mathbf{W}_j; \mathbf{Z}_{i,j})$ is the same for all $j \in [k]$ which is equal to $I(\mathbf{W}; \mathbf{Z}_i)$. Therefore,

$$
I(\mathbf{W}^*, \mathbf{J}, \mathbf{R}; \mathbf{Z}_i^k) \leq kI(\mathbf{W}; \mathbf{Z}_i) + \log(2k). \tag{46}
$$

Combining (44–46) gives

$$
\mathbb{E}\left[\max_{j \in [k]} \left| L_\mu(\mathbf{W}_j) - L_{\mathbf{S}_j}(\mathbf{W}_j) \right| \right] \leq \frac{1}{n} \sum_{i=1}^{n} \sqrt{2\sigma^2 \left(kI(\mathbf{W}; \mathbf{Z}_i) + \log(2k)\right)}. \tag{47}
$$

Since $(\mathbf{S}_j, \mathbf{W}_j)$ with $j \in [k]$ are independent copies of $(\mathbf{S}, \mathbf{W})$, then for any $\alpha > 0$,

$$
\Pr\left(\max_{j \in [k]} \left| L_\mu(\mathbf{W}_j) - L_{\mathbf{S}_j}(\mathbf{W}_j) \right| < \alpha \right) = \Pr\left( \left| L_\mu(\mathbf{W}) - L_{\mathbf{S}}(\mathbf{W}) \right| < \alpha \right)^k. \tag{48}
$$

By Markov's inequality,

$$
\Pr\left(\max_{j \in [k]} \left| L_\mu(\mathbf{W}_j) - L_{\mathbf{S}_j}(\mathbf{W}_j) \right| \geq \alpha \right) \leq \frac{1}{\alpha} \mathbb{E}\left[\max_{j \in [k]} \left| L_\mu(\mathbf{W}_j) - L_{\mathbf{S}_j}(\mathbf{W}_j) \right| \right]. \tag{49}
$$

Substituting (47), (48) into (49) leads to

$$
1 - \Pr\left( \left| L_\mu(\mathbf{W}) - L_{\mathbf{S}}(\mathbf{W}) \right| < \alpha \right)^k \leq \frac{1}{\alpha n} \sum_{i=1}^{n} \sqrt{2\sigma^2 \left(kI(\mathbf{W}; \mathbf{Z}_i) + \log(2k)\right)},
$$

which is equivalent to

$$
\Pr\left( \left| L_\mu(\mathbf{W}) - L_{\mathbf{S}}(\mathbf{W}) \right| < \alpha \right) \geq \left(1 - \frac{1}{\alpha n} \sum_{i=1}^{n} \sqrt{2\sigma^2 \left(kI(\mathbf{W}; \mathbf{Z}_i) + \log(2k)\right)} \right)^{1/k}. \tag{50}
$$

For any given $\delta \in (0, 1)$, we take $k = \lfloor 1/\delta \rfloor$ and

$$
\alpha^* = \frac{2}{n} \sum_{i=1}^{n} \sqrt{2\sigma^2 \left(\frac{I(\mathbf{W}; \mathbf{Z}_i)}{\delta} + \log \frac{2}{\delta}\right)}.
$$

Hence, (50) indicates that

$$
\begin{aligned}
\Pr\left( \left| L_\mu(\mathbf{W}) - L_{\mathbf{S}}(\mathbf{W}) \right| < \alpha^* \right) &\geq \left(1 - \frac{1}{\alpha^* n} \sum_{i=1}^{n} \sqrt{2\sigma^2 \left(\frac{I(\mathbf{W}; \mathbf{Z}_i)}{\delta} + \log \frac{2}{\delta}\right)} \right)^{1/\lfloor 1/\delta \rfloor} \\
&= \frac{1}{2}^{1/\lfloor 1/\delta \rfloor} \geq 1 - \delta.
\end{aligned}
$$

$\square$

# B   Additional Experiments

We conduct additional numerical experiments on CIFAR-10 (Krizhevsky et al., 2009) to further validate our generalization bound in Theorem 1 (see Figure 3, 4 and Table 2).

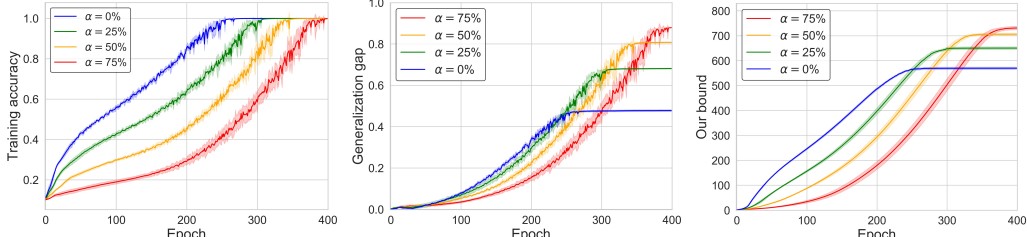

Figure 3: Illustration of our generalization bound in Theorem 1. We use the SGLD algorithm to train convolutional neural networks (CNNs) on CIFAR-10 when the training data have different label corruption level $\alpha \in \{0\%, 25\%, 50\%, 75\%\}$. Left: training accuracy. Middle: (empirical) generalization gap. Right: (empirical) generalization bound.

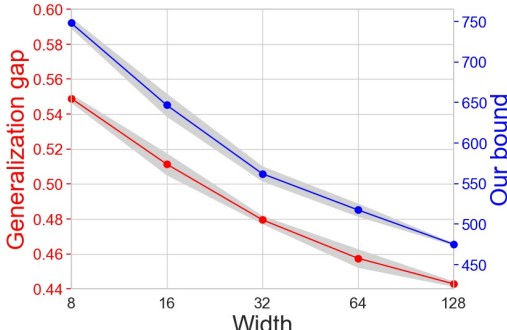

Figure 4: Comparison between the generalization gap and our generalization bound in Theorem 1. We use the SGLD algorithm to train CNNs with varying widths (i.e., number of filters in CNN) on CIFAR-10. As shown, both the generalization gap and our bound are decreasing w.r.t. the network width.

| dataset | method | lr | width | depth | $\tau$ | $\Psi$ | MI |
|---------|--------|-----|-------|-------|--------|--------|-----|
| CIFAR-10 | OURS (THEOREM 1) | **0.41** | **0.93** | **1.00** | **0.45** | **0.78** | **0.25** |
| | NEGREA ET AL. (2019) | 0.33 | 0.41 | 0.85 | 0.38 | 0.53 | 0.16 |

Table 2: We adopt the three evaluation criteria proposed in Jiang et al. (2019) for comparing our generalization bound with the benchmark method (Negrea et al., 2019): (i) Kendall's rank-correlation coefficient ($\tau$), (ii) Granulated Kendall's coefficient ($\Psi$), and (iii) conditional independent test (MI). All scores, except MI, are within $[-1, 1]$ and the score of MI is normalized to $[0, 1]$. We also report the correlations when a single hyper-parameter (e.g., learning rate (lr)) is varying.

| Parameter | Details |
|---|---|
| Dataset | MNIST |
| Number of training data | 5000 |
| Batch size | 500 |
| Learning rate | Initialization = 0.03, decay rate = 0.96, decay steps=2000 |
| Inverse temperature | $\beta_t = 10^6/(2\eta_t)$ |
| Architecture | MLP with ReLU activation |
| Depth | 3 layers |
| Width | 64 hidden units |
| Objective function | Cross-entropy loss |
| Loss function | 0-1 loss |

Table 3: Experiment details of Figure 1, 2 and Table 1. For Figure 2, the network width is varying among $\{16, 32, 64, 128, 256\}$ hidden units. For Table 1, we run the SGLD algorithm 600 epochs and vary three hyper-parameters: learning rate initialization $\in \{0.03, 0.06, 0.09\}$, depth $\in \{2, 3, 4\}$, and width $\in \{16, 32, 64\}$.

## C Supporting Experimental Results

Recall that our generalization bound in Theorem 1 involves the variance of gradients. To estimate this quantity from data, we repeat our experiments 4 times and record the batch gradient at each iteration. This batch gradient is the one used for updating the parameters in the SGLD algorithm so it does not require any additional computations. Then we estimate the variance of gradients by using the population variance of the recorded batch gradients. Finally, we repeat the above procedure 4 times for computing the standard deviation, leading to e.g., the shaded areas in Figure 1. We provide experimental details in Table 3 and 4 for reproducing our experiments.

| Parameter | Details |
|---|---|
| Dataset | CIFAR-10 |
| Number of training data | 5000 |
| Batch size | 500 |
| Number of epochs | 400 |
| Learning rate | Initialization = 0.03, decay rate = 0.96, decay steps = 2000 |
| Inverse temperature | $\beta_t = 10^6/(2\eta_t)$ |
| Architecture | $\mathsf{conv}(5,32)$ $\mathsf{pool}(2)$ $\mathsf{conv}(5,32)$ $\mathsf{pool}(2)$ $\mathsf{fc}(120)$ $\mathsf{fc}(84)$ $\mathsf{fc}(10)$ |
| Objective function | Cross-entropy loss |
| Loss function | 0-1 loss |

Table 4: Experiment details of Figure 3, 4 and Table 2. Here $\mathsf{conv}(k,w)$ is a $k \times k$ convolutional layer with $w$ filters; $\mathsf{pool}(k)$ is a $k \times k$ max pooling layer; and $\mathsf{fc}(k)$ is a fully connected layer with $k$ units. The convolutional layers and the fully connected layers all use ReLU activation function. For Figure 4, the network width (i.e., number of filters in CNN) is varying among $\{8, 16, 32, 64, 128\}$. For Table 2, we are varying three hyper-parameters: learning rate initialization $\in \{0.03, 0.06, 0.09\}$, number of convolutional layers $\in \{2, 3, 4\}$, and width $\in \{32, 64, 128\}$.