# OpenReview forum: "Analyzing the Generalization Capability of SGLD Using Properties of Gaussian Channels"
_NeurIPS.cc/2021/Conference — NeurIPS 2021 Poster_

### Official Review · Reviewer_HZ1y · 2021-07-14

**Rating:** 6
**Confidence:** 4

**Summary:**

This paper gives a generalization bound for the stochastic gradient Langevin dynamics (SGLD) algorithm using tools from information theory (notably, the SDPI for Gaussian channels and the HWI inequality). The bound is subsequently used to analyze generalization of differentially private SGD (DP-SGD). Supporting experiments on the CIFAR-10 dataset show that the bound is more favorably correlated with the generalization error than prior work by Negrea et al. (2019).


**Limitations And Societal Impact:**

The authors claim that that is no direct negative societal impact of their work and I agree.


**Main Review:**

*** Post-rebuttal ***

I am increasing my score to 6 based on the authors' response.

***

Generalization bounds for SGLD have recently attracted a lot of attention in the NeurIPS community and the contribution is timely.

Originality:

-- The analysis primarily builds on the works of Xu and Raginsky (2017), and Pensia et al. (2018) that bound the expected generalization error in terms of the mutual information (MI) between the input training data and the output hypothesis under a sub-Gaussian loss. Since the date generating distribution is unknown, estimating the mutual information is intractable. This issue is addressed by upper-bounding the MI. The bounds thus obtained (for DP-SGD) are expressed as a sum of errors associated with each training iteration and features the variance of the gradients (from upper bounding the MI + chain rule of MI) and a time-decaying factor (by way of Gaussian SDPI) that weighs the contributions from early iterations of SGLD. The intuition for the time-decaying factor is natural as it factors in the cumulative effect of noise.

-- In terms of originality, application of the SDPI and HWI inequality for Gaussians appear novel.


Issues:

-- It appears that there is a version of this work (arXiv:2102.02976v1, dated Feb 5th, 2021) with a different title ``"Learning While Dissipating Information: Understanding the Generalization Capability of SGLD."  I will recommend that the authors cite this work. Also a question: Theorem 1 in the arXiv version apparently gives a bound for SGLD (sampling with replacement) that features the time-decay factor. I believe the corresponding bound in the present submission is Theorem 2, which does NOT feature the time-decay factor. If the latter is a correction of sorts for the one in the arXiv version, then this should be stated clearly with citation.

-- I can understand that the DP-SGD is a special case of SGLD, but the motivation for selecting this specific algorithm is not very clear. Ideally, the general version of the algorithm (SGLD, Theorem 2) should be stated before stating a special case (DP-SGD, Theorem 1). However, the order is reversed here. I believe this was done to accentuate the role of the time-decay factor. In any case, the organization and overall presentation of the paper can be significantly improved.


Suggestion (minor):

-- The isotropic noise structure of SGLD is integral to the proof here and I am inclined to believe that the technique cannot be applied to SGD. Nonetheless, a discussion on if and how any of the proposed techniques can be used to analyze SGD might benefit the potential reader.

**Time Spent Reviewing:**

Three

---

> ### Author Response · Authors · 2021-08-09
> **Response to the comments and concerns from Reviewer HZ1y**
>
> We thank the reviewer for providing comments and regarding our proof techniques as novel!
>
> $\textbf{[Arxiv version]}$
>
> The reason that Theorem 2 does not have a decay factor is because we consider a more general setting in the present submission: the final output from SGLD can be any function of the parameters across all iterations (i.e., $W = f(W_1, …, W_T)$ for an arbitrary function $f$) instead of the parameter at the last iteration (i.e., $W = W_T$) considered in arXiv:2102.02976v1. We understand that this difference is very subtle and is likely to be overlooked. However, the proof of Theorem 1 in arXiv:2102.02976v1 does not hold under this general setting. In fact, we can even take a step forward and prove that it could be *impossible* to incorporate a decay factor under this general setting.
>
> >Consider an extreme scenario in which the SGLD algorithm outputs the parameters across all iterations (i.e., $W=(W_1,...,W_T)$). The data processing inequality yields that for a data point $Z_i$ used at the $t$-th iteration, $I(W;Z_i) \geq I(W_t;Z_t) > 0$. Hence, it is impossible to have $I(W;Z_i) \to 0$ as $T\to \infty$ in contrast with Lemma 4.
>
> Hence, we developed a completely different approach for proving Theorem 2 and believe that this generalization is worthy because many existing theoretical analyses of SGLD (e.g., Zhang et al., 2017; Jin et al., 2019) require the final output from SGLD to be the parameter which achieves the smallest value of the loss function or is closest to a stationary point, but not necessarily $W_T$.
>
> $\textbf{[DP-SGD]}$
>
> The DP-SGD algorithm is an application in which the decay factor appears and is often non-trivial (i.e., it is strictly upper bounded by 1). In fact, since our decay factor relies on a uniform upper bound of the gradient norm, denoted by $K$, this $K$ can be large in general, leading to a decay factor close to $1$. However, in the setting of DP-SGD, since the gradient is clipped for controlling the sensitivity of each update, this $K$ is not very large.
>
> We like the reviewer’s suggestion regarding the organization of Section 3 (i.e., present a general result first and apply it to DP-SGD) and will implement it in the revised paper! We also plan to revise our Theorem 1 as follows.
>
> >First, we recall an implementation of the DP-SGD algorithm [see e.g., Algorithm 1 in Feldman et al., 2018]. The model parameters are initialized at a random point $W_0 \in \mathcal{W}$ and updated using the following rule:
> $$W_t = \mathsf{Proj}(W_{t-1} - \eta(\mathsf{Clip}(\nabla \ell_w(W_{t-1}, Z_t)) + N))$$
> where $Z_t$ is drawn from the dataset without replacement and for a constant $K>0$ the clip operator is defined as
> $$\mathsf{Clip}(w) = \frac{K}{\max(K, \|w\|)} w $$
> The above recursion is run $T$ iterations and the final output is $W_T$.
>
> >[New Theorem 1] Under Assumption 1 and the above setup, the expected generalization gap can be upper bounded by
> $$\frac{2\sigma}{n} \sum_{t=1}^T \sqrt{\mathsf{Var}\left(\mathsf{Clip}\left(\nabla_{w} \ell(W_{t-1}, Z\right)\right)) \cdot q^{T-t}}$$
> where $q = 1 - 2\bar{\Phi}\left((D+2\eta K)/2\eta\right) \in [0,1]$.
>
> >[Proof] The proof follows directly by replacing the gradient term in the proof of Theorem 1 with the clipped gradient.
>
>
> $\textbf{[Suggestion]}$
>
> You are correct: our proof technique relies on an explicit additive noise in the SGLD algorithm. Although we believe that our framework can be used to analyze other noisy iterative algorithms, it cannot be applied to the SGD algorithm directly. As you suggested, we will include a discussion on limitation/future work in our revised paper.
>
> We hope that the above discussions can address all your concerns!

---

### Official Review · Reviewer_QtYi · 2021-07-17

**Rating:** 6
**Confidence:** 3

**Summary:**

This paper derives upper bounds on the generalization error of SGLD and DP-SGD using an information theoretic analysis. Compared to past work on this problem the authors derive upper bounds that depend on the sum of variance of the gradients along the path, converge at a rate of 1/n, and discount the effect of samples seen early in the training run.

**Limitations And Societal Impact:**

I do not think there is any potential negative societal impact of this work.

**Main Review:**

As mentioned by the authors, previous papers have used information theoretic tools to obtain generalization bounds that have one or more of the following properties: a) upper bounds that depend on the sum of variance of the gradients along the path or on the expected norm of the gradients, b) converge at a rate of 1/n, and c) discount the effect of samples seen early in the training run; but none of the previous bounds have satisfy all of the properties simultaneously. The authors derive such a bound, and the main technical driving force for their analysis seems to be the use of strong data processing inequalities. While I appreciate this advance, it does seem like a narrow, technical step forward to me in light of the past work on this area, including the paper of Xu and Raginsky, who also previously suggested using strong data processing inequalities to get tighter bounds. Other comments and questions about the paper are listed below:

1. The abstract suggest that SGLD is a widely used optimizer, but the authors do not provide any citations to support this claim. The authors should back up this claim with citations in the text.
2. Do the bounds of Li et al. (2019) discussed in ln 94 also depend on the sum of the variance of the gradients or on the expected norms of the gradients?
3. Ln. 219 says that DP-SGD is a special instance of SGLD algorithm. I am not sure why this is the case, since the gradients are clipped in the DP-SGD algorithm (see Algorithm 1, Abadi et al. (2016)). How does the theory handle this clipping? Relatedly, do Theorems 1 and 2, and Corollary 1 apply to the DP-SGD algorithm? The authors should clarify this.
4. In ln. 246, if I understand correctly, the sharpness studied Jiang et al. 2019 refers to the robustness of the network's output when the weights are perturbed slightly. Why does the variance of the gradient relate to the sharpness studied there?
5. Do the authors expect a sharper version of Theorem 2 to hold if the average iterate is returned instead of an arbitrary function of the weights.
6. Regarding the experiments, in Table 1, I am not quite sure why the authors only compare their generalization bounds to that of Negrea et al. The authors say that the previous papers Mou et al., Pensia et al. ,... did not perform a empirical study, but Table 1 only compares the correlation between the actual generalization performance of a network and the theoretical bound. I believe the authors should add these comparisons to their empirical study.



============== Post author rebuttal =============
Thank you for carefully answering my questions. I have raised my score as a result.

**Time Spent Reviewing:**

2

---

> ### Author Response · Authors · 2021-08-09
> **Response to the comments and concerns from Reviewer QtYi**
>
> We thank the reviewer for the comments! First, we would like to highlight our following technical contributions which were not reflected in the initial reviews.
>
> (a) Although leveraging strong data processing inequalities for tightening bounds sounds intuitive, using it to incorporate a decay factor is highly non-trivial. The challenge is that one must use the internal additive noise in the SGLD algorithm instead of relying on external noise added to the final output from the algorithm, as suggested by e.g., Xu and Raginsky. In fact, the latter idea (i.e., post-processing the output by adding external noise) can only tighten the bound by a factor of $q_T$ while our bound has an exponential decay factor $\prod_{t'=t+1}^T q_{t'}$. This is also why none of the existing bounds by [Pensia et al., 2018; Bu et al., 2020; Negrea et al., 2019] have a decay factor.
>
> (b) Since estimating $I(W_t; Z_i)$ is intractable in practice, we derive an upper bound using the variance of gradients in Lemma 5. The proof is complex as it relies on Lemma 2 and a careful construction of a coupling distribution for upper bounding the Wasserstein distance. In fact, we note that [Negrea et al., 2019, Neu 2021] have attempted to develop a generalization bound depending on the variance of gradients but their variance-based bounds only have a weak order $O(1/\sqrt{n})$ in terms of the sample size $n$ while ours is $O(1/n)$.
>
> (c) In order to extend our framework to a more general setting in which the SGLD output can be any function of the parameters across all iterations (i.e., $W = f(W_1, …, W_T)$) rather than the parameter at the last iteration (i.e., $W = W_T$), we develop a completely different proof technique in Theorem 2 by using the chain rule of mutual information and properties of Gaussian channels.
>
> We kindly ask the reviewer to take into account all the above contributions as well.
>
> Regarding your other comments,
>
> 1.  Yes, we will provide more references and add the following discussion to the revised paper.
>
> >From a theoretical perspective, the SGLD, as well as other noisy iterative algorithms, could escape local minima [Kleinberg et al., 2018] and saddle points [Ge et al., 2015]. From a practical consideration, SGLD algorithm is often used for training machine learning (ML) models while protecting user privacy. For example, Opacus and TensorFlow privacy have been recently announced for training networks with differential privacy guarantees. The additive noise in SGLD may also facilitate learning for DNNs [Neelakantan et al., 2015].
>
> 2. The SGLD bound in Li et al. (2019) relies on an expected squared gradient norm instead of variance of gradients (see their Theorem 11).
>
> 3. This is a great point! To answer your question, we will revise our Theorem 1 and include the following result.
>
> >First, we recall an implementation of the DP-SGD algorithm [see e.g., Algorithm 1 in Feldman et al., 2018]. The model parameters are initialized at a random point $W_0 \in \mathcal{W}$ and updated using the following rule:
> $$W_t = \mathsf{Proj}(W_{t-1} - \eta(\mathsf{Clip}(\nabla \ell_w(W_{t-1}, Z_t)) + N))$$
> where $Z_t$ is drawn from the dataset without replacement and for a constant $K>0$ the clip operator is defined as
> $$\mathsf{Clip}(w) = \frac{K}{\max(K, \|w\|)} w $$
> The above recursion is run $T$ iterations and the final output is $W_T$.
>
> >[New Theorem 1] Under Assumption 1 and the above setup, the expected generalization gap can be upper bounded by
> $$\frac{2\sigma}{n} \sum_{t=1}^T \sqrt{\mathsf{Var}\left(\mathsf{Clip}\left(\nabla_{w} \ell(W_{t-1}, Z\right)\right)) \cdot q^{T-t}}$$
> where $q = 1 - 2\bar{\Phi}\left((D+2\eta K)/2\eta\right) \in [0,1]$.
>
> >[Proof] The proof follows directly by replacing gradient term in the proof of Theorem 1 with the clipped gradient.
>
> 4. Yes, your understanding is correct. However, Jiang et al. 2019 also proposed some potential optimization-based measures in their Section 4.4. In particular, they observed through experiments that "Towards the end of the training, the variance of the gradients also captures a particular type of “flatness” of the local minima."
>
> 5. An advantage of our Theorem 2 is that it provides a generalization bound for an arbitrary function of the weights. In contrast, many existing approaches can only analyze the weight at the last iteration. For some special functions (e.g., $W=W_t$ with $t<T$), it is possible to get a sharper bound by adapting one line of our proof (Eq. (27) in Appendix). Nonetheless, for the average iterate, we are unsure if the bound can be tighter.
>
> 6. We take the bound in Negrea et al., 2019 as a benchmark because it is the current state-of-the-art generalization bound for the SGLD algorithm. It has a similar form with our bound and the order is comparable with ours. The bound in Pensia et al., 2018 involves a Lipschitz constant and it is unclear to us how this constant can be computed. Compared with Mou et al., 2018, in theory, our bound is already sharper ($O(1/n)$ vs. $O(1/\sqrt{n})$) in terms of the dependence of sample size $n$.
>
> We hope that the above discussions can address all your concerns and will include them in our revised paper.
>
> References:
>
> 1. Kleinberg, B., Li, Y. and Yuan, Y., 2018, July. An alternative view: When does SGD escape local minima?. In International Conference on Machine Learning (pp. 2698-2707). PMLR.
>
> 2. Ge, R., Huang, F., Jin, C. and Yuan, Y., 2015, June. Escaping from saddle points—online stochastic gradient for tensor decomposition. In Conference on learning theory (pp. 797-842). PMLR.
>
> 3. Neelakantan, A., Vilnis, L., Le, Q.V., Sutskever, I., Kaiser, L., Kurach, K. and Martens, J., 2015. Adding gradient noise improves learning for very deep networks. arXiv preprint arXiv:1511.06807.

---

### Official Review · Reviewer_kujL · 2021-07-19

**Rating:** 6
**Confidence:** 4

**Summary:**

This paper presents some new results on bounding the generalization error of the SGLD algorithm using the properties of Gaussian channels. For SGLD without sample replacement, strong data processing inequality for Gaussian channel is used to show a time-wise contraction of the upper bound. For the general SGLD with sample replacement, a result that relates the mutual information of Gaussian channel to the input variance is used to obtain a more general upper bound.
Experiments are also performed to show some interesting correlations between the obtained upper bounds and the true generalization error.

**Limitations And Societal Impact:**

The major limitations are the lack of discussions on the obtained bounds and missing details of the experimental results. But I think the paper can be easily improved by addressing these limitations.

**Main Review:**

While the results presented in the paper look to be new and the experimental results look interesting, there are some aspects that could be improved.
1. The term "DP SGD" is mentioned several times in the paper without an explicit statement. It looks to be referring to SGD without sample replacement, but it would be better if this can be explicitly stated.
2. For Theorem 1, obtaining a time-decaying (contraction) is good, but there is no further discussion on this factor, e.g. how does the final bound scale as T increases. Even some qualitative discussion on this factor would be useful to assess its significance.
3. For Theorem 2, it is claimed that the final bound scales in n as O(1/n). However, the actual dependence of the bound on n may not be that simple, because here n = m*b, which means that as n increases, both m and b would increase. It is thus not clear whether the O(1/n) can be maintained as n increases. Also, at least some qualitative discussion on the dependence on n and T would be helpful.
4. For the experimental results, many details are missing. For example, how are the variance terms in the bound (Theorem 2) evaluated?
How are the bounds scaled to match the empirical generalization error?
How is the correlation between the bound the empirical generalization defined?
It can also be seen that the empirical generalization error in all cases saturates as T increases. Is this predicted by the bound? This is another reason why more discussions on the bounds themselves should be provided.

**Time Spent Reviewing:**

5

---

> ### Author Response · Authors · 2021-08-09
> **Response to the comments and concerns from Reviewer kujL**
>
> We thank the reviewer for the kind comments and the encouragement!
>
> 1. I agree that providing an explicit statement can help the readers understand the results better. To address your concern, we will revise our Theorem 1 and include the following discussion in the paper.
>
> >First, we recall an implementation of the DP-SGD algorithm [see e.g., Algorithm 1 in Feldman et al., 2018]. The model parameters are initialized at a random point $W_0 \in \mathcal{W}$ and updated using the following rule:
> $$W_t = \mathsf{Proj}(W_{t-1} - \eta(\mathsf{Clip}(\nabla \ell_w(W_{t-1}, Z_t)) + N))$$
> where $Z_t$ is drawn from the dataset without replacement and for a constant $K>0$ the clip operator is defined as
> $$\mathsf{Clip}(w) = \frac{K}{\max(K, \|w\|)} w $$
> The above recursion is run $T$ iterations and the final output is $W_T$.
>
> >[New Theorem 1] Under Assumption 1 and the above setup, the expected generalization gap can be upper bounded by
> $$\frac{2\sigma}{n} \sum_{t=1}^T \sqrt{\mathsf{Var}\left(\mathsf{Clip}\left(\nabla_{w} \ell(W_{t-1}, Z\right)\right)) \cdot q^{T-t}}$$
> where $q = 1 - 2\bar{\Phi}\left((D+2\eta K)/2\eta\right) \in [0,1]$.
>
> >[Proof] The proof follows directly by replacing gradient term in the proof of Theorem 1 with the clipped gradient.
>
> 2. This is a great suggestion! We will include the following discussion on the role of the decay factor after Theorem 1.
>
> >The contribution from each iteration to our bound is composed of a time-decay factor and a variance term. As the number of iterations increases, the variance term becomes negligible, especially when the algorithm converges to a minimum. Hence, without the decay factor, the bound is determined by the early iterations before the algorithm converges. This is counterintuitive, since one may expect that the later iterations have a higher impact on generalization as they more closely determine the output $W_T$. The role of the decay factor is to complement this issue and to enable the contributions from early iterations to reduce.
>
> >To qualitatively analyze the role of the decay factor, we demonstrate that as $T\to \infty$, the generalization bound in Theorem 1 converges to 0, whereas the counterpart without a decay factor may tend to a large constant. We focus on the role of the decay factor by choosing a constant step size and inverse temperature (i.e., $\beta_t =\beta$ and $\eta_t =\eta$ for all $t$) and upper bounding the variance term by the Lipschitz constant $K$. In this case, each component $q_t$ in the decay factor is invariant w.r.t. $t$ (i.e., $q_t = q < 1$ for all $t$). Hence, our bound in Theorem 1 becomes
> $$\frac{C}{n} \sum_{t=1}^T \sqrt{q^{T-t}}$$
> where the constant $C = \sqrt{2\sigma^2 \beta \eta K}$. This bound can be further simplified as
> $$\frac{C}{n} \frac{1-q^{T/2}}{1-\sqrt{q}}$$
> As $T \to \infty$ (and $n\to \infty$ as well since we consider sampling without replacement), the above bound clearly converges to $0$. However, without the decay factor, the corresponding bound can only tend to the constant $C>0$. This example illustrates the importance of having a decay factor in the bound.
>
> 3. To answer your question, we will revise our related work section and include the following more precise discussion after Corollary 1.
>
> >The order of our generalization bound in Corollary 1 is
> $$\min\left(\frac{1}{n} \sum_{t=1}^T \sqrt{\beta \eta_t}, \sqrt{\frac{\beta}{bn} \sum_{t=1}^T \eta_t} \right).$$ Our bound is distribution-dependent through the variance of gradients in contrast with [Pensia et al., 2018; Bu et al., 2020; stability bound in Mou et al., 2018]. It is also tighter than [PAC-Bayesian bound in Mou et al., 2018] whose order is $\sqrt{\frac{\beta}{n} \sum_{t=1}^T \eta_t}$. Our bound is applicable regardless of the choice of learning rate while the bound in [Li et al., 2019] requires the scale of the learning rate to be upper bounded by the reciprocal of the Lipschitz constant which, as already pointed out by [Negrea et al., 2019], limits its applicability to typical learning problems since the Lipschitz constant can be prohibitively large. Our Corollary 1 has the same order with [Negrea et al., 2019] but we incorporate an additional decay factor under additional assumptions and numerical experiments suggest that our bound is more favourably correlated with the true generalization gap. Finally, we remark that the purpose of introducing Corollary 1 is to derive a generalization bound that has a similar form with existing bounds so that we can compare. However, our main result (Theorem 2) is often tighter than Corollary 1, since this corollary is obtained by combining Theorem 2 with Minkowski inequality and Jensen's inequality.
>
>
> 4. We thank the reviewer for pointing out this issue and will provide the following experimental details in our paper.
>
> >We repeat our experiment 4 times and record the batch gradient at each iteration. This batch gradient is the one used for updating the parameters in the SGLD algorithm so it does not require any additional computations. Then we estimate the variance of gradients in Theorem 2 by using the population variance of the recorded batch gradients. Finally, we repeat the above procedure 4 times for computing the standard deviation, leading to the shaded areas in Figure 1.
>
> >To evaluate the correlation between the bound and the generalization gap, we use the three evaluation criteria introduced in [Jiang et al., 2019 Section 2.2]. Take Kendall’s rank-correlation coefficient as an example. It measures whether the bound and the generalization gap are increase (or decreasing) simultaneously if hyper-parameters (e.g., learning rate) vary.
>
> >The generalization gap tends to be stable since the algorithm converges to a minimum. This can be seen from Figure 1 Left. Our generalization bound captures this phenomenon as the variance of gradients becomes negligible when the algorithm starts converging. The intuition is that the variance of gradients reflects a certain kind of sharpness of the loss landscape and as the algorithm converges to a minimum, the loss landscape becomes flatter.
>
> We hope that the above discussions can answer all your concerns and will include them in our revised paper.

---

> > ### Comment · Reviewer_kujL · 2021-08-31
> > **Thanks for the response**
> >
> > Thanks to the authors for the response. After reading it, the experimental methods are clearer, but I still have some concern on the significance of the paper.
> > A main concern is the claim of the O(1/n) of the dependence of the generalization bound on n.
> >
> > 1. For Theorem 1, it looks like the DP-SGD is essentially an SGLD with minibatch size of 1, and with the sample size = number of iterations, that is n = T. First, this case is not practical, as it assumes that there is always a fresh sample provided in each iteration, while in practical model training, the data is usually reused by multiple training epochs.  Second, in this case, under proper smoothness conditions on the model and the loss
> > function, isn't it well-known that even plain SGD can achieve a generalization error of O(sqrt(1/T)) or even O(1/T)? If so, what is the significance of Theorem 1?
> >
> > 2. For Theorem 2, if b is fixed, then m would grow linearly as n in the stated bound. Then the dependence of O(1/n) can hardly hold. For Corollary 1, which is a bound in terms of n and T, it looks like when T is fixed, the bound is inversely proportional to n. If this is the main message, why there is no experiments supporting this result? It seems that all experiments are done to reflect the dependence of the generalization error on T, instead of n. Also, there is no specification or discussion on how the data is actually sampled for a given T, and how that would affect the bound.
> > This is the reason why I have concerns on the significance of the results, if I didn't miss anything.

---

> > > ### Author Response · Authors · 2021-09-01
> > > **Thanks for the follow-up comments!**
> > >
> > > Thanks for the follow-up comments! We are glad to hear that the experimental details are clear now.
> > >
> > > 1. Theorem 1
> > >
> > > [$\textbf{Significance}$] To clarify, Theorem 1 is not a special instance of Theorem 2. As a reminder, Theorem 2 derives a SGLD generalization bound which allows the output $W$ to be any function of the parameters across all iterations (i.e., $W=f(W_1,\cdots,W_T)$). On the other hand, Theorem 1 proves that if the output is the parameter at the last step (i.e., $W=W_T$), one can further tighten the bound by incorporating a time-decaying factor. Please recall that the importance of the decay factor has been qualitatively demonstrated through an example in our previous response to your second concern. From a technical perspective, the proof of Theorem 1 is completely different from Theorem 2 in order to reveal the decay factor.
> > >
> > > [$\textbf{Motivation}$] Our analysis is motivated by a recent line of works on privacy amplification by iteration, originally proposed by [Feldman et al., 2018]. Specifically, [Feldman et al., 2018] provided two intertwined observations of the DP-SGD algorithm: (i) not releasing the intermediate steps can amplify the privacy guarantees and (ii) data points used in the early iterations enjoy stronger privacy than those occurring late. In this paper, we establish two analogous results through different proof techniques: (i) our generalization bound can be sharpened by incorporating a time-decaying factor if DP-SGD only outputs the parameter at the last step (Theorem 1) and (ii) this decay factor enables the impact of early iterations on our bound to reduce with time (Lemma 4).
> > >
> > > [$\textbf{Novelty}$] A standard approach [see e.g., He et al., 2020] of deriving a generalization bound for DP-SGD algorithm follows two steps: (i) prove that DP-SGD satisfies the $(\epsilon,\delta)$-DP guarantees; (ii) derive/apply a generalization bound that holds for *any* $(\epsilon,\delta)$-DP algorithm. However, generalization bounds obtained from this procedure are distribution-independent since DP is robust with respect to data distribution. In contrast, our bound in Theorem 1 is distribution-dependent through the variance term. Finally, our bounds require mild assumptions and are applicable even when the loss function is non-convex and non-Lipschitz continuous. In contrast, most existing generalization bounds for SGD and its variants rely on stronger assumptions.
> > >
> > > [$\textbf{Clarification}$] Sampling without replacement is a common assumption in the literature of DP-SGD [see e.g., Song et al., 2013; Wu et al., 2017; Feldman et al., 2018; Balle et al., 2019; Asoodeh et al., 2020]. We also get rid of this assumption in Theorem 2 by leveraging the chain rule of mutual information.
> > >
> > > 2. Theorem 2
> > >
> > > [$\textbf{1/n order}$] First, when we discuss the order of sample size dependence, we assume that the total number of iterations $T$ is fixed. Your interpretation of "if $b$ is fixed, then $m$ would grow linearly as $n$" is correct! However, since $T$ is fixed, the inner summation of $t \in T_j$ in our bound will include fewer terms as $n$ grows. We believe the best way to understand the order of Theorem 2 is by further upper bounding Theorem 2 (this is why we include Corollary 1). Specifically, since $\sqrt{\sum_i x_i}\leq \sum_i \sqrt{x_i}$, we can move the second summation in our bound outside the square root, leading to the following upper bound of Theorem 2:
> > > $$\frac{\sqrt{2b}\sigma}{2n} \sum_{i=1}^m \sum_{t\in T_i} \sqrt{\beta_t \eta_t \mathsf{Var}(...)}.$$
> > > This expression is equivalent to
> > > $$\frac{\sqrt{2b}\sigma}{2n} \sum_{t=1}^T \sqrt{\beta_t \eta_t \mathsf{Var}(...)}.$$
> > > As we fix $T$, when the summands over $t$ are bounded (which holds when $\beta_t\eta_t$ and $\mathsf{Var}$ are bounded), the above quantity has $O(1/n)$ dependence on the number of samples. Since our Theorem 2 is always tighter than the above quantity, its order is no worse than $O(1/n)$. Finally, we remark that we can even get rid of the dependence of $\sqrt{b}$ in the numerator of the above quantity (see the proof of Corollary 1 for a detailed discussion).
> > >
> > > [$\textbf{Main message and Experiment}$] The main reason we discuss the $O(1/n)$ dependence on the sample size is to compare with existing SGLD generalization bounds since some of them [e.g., Theorem 2 of Mou et al., 2018] can only achieve a weaker order $O(1/\sqrt{n})$. Nonetheless, the goal of this paper is to study generalization phenomena through our bounds. In particular, since SGLD is an iterative algorithm, we would like to understand how each iteration impacts the algorithmic generalization capability. This is why we illustrate our bound on different values of $T$ in Figure 1; so do many other papers on SGLD generalization bounds [Negrea et al., 2019; Li et al., 2019].
> > >
> > > Regarding how we sample data for a given $T$, we are not sampling data for each $T$ independently. Instead, we first randomly select 5k training data and split them into mini-batches. These mini-batches are fed into the algorithm in sequence in each epoch. We run the SGLD algorithm 1400 epochs for MNIST and record the batch gradient at each iteration. Finally, we compute our bound for $T$ from the first to the last epoch by using the recorded gradient information.
> > >
> > > We hope the above discussions can answer all your concerns and would like to thank you again for your time, effort, and thoughtful feedback!
> > >
> > > $\textbf{References}$
> > >
> > > 1. Feldman, V., Mironov, I., Talwar, K. and Thakurta, A., 2018, October. Privacy amplification by iteration. In IEEE 59th Annual Symposium on Foundations of Computer Science.
> > >
> > > 2. Asoodeh, S., Diaz, M. and Calmon, F.P., 2020. Privacy Analysis of Online Learning Algorithms via Contraction Coefficients. In International Symposium on Information Theory.
> > >
> > > 3. He, F., Wang, B. and Tao, D., 2020. Tighter generalization bounds for iterative differentially private learning algorithms. arXiv preprint arXiv:2007.09371.
> > >
> > > 4. Song, S., Chaudhuri, K. and Sarwate, A.D., 2013, December. Stochastic gradient descent with differentially private updates. In 2013 IEEE Global Conference on Signal and Information Processing.
> > >
> > > 5. Balle, B., Barthe, G., Gaboardi, M. and Geumlek, J., 2019. Privacy amplification by mixing and diffusion mechanisms. arXiv preprint arXiv:1905.12264.
> > >
> > > 6. Wu, X., Li, F., Kumar, A., Chaudhuri, K., Jha, S. and Naughton, J., 2017, May. Bolt-on differential privacy for scalable stochastic gradient descent-based analytics. In 2017 ACM International Conference on Management of Data.
> > >
> > > 7. Mou, W., Wang, L., Zhai, X. and Zheng, K., 2018, July. Generalization bounds of sgld for non-convex learning: Two theoretical viewpoints. In Conference on Learning Theory.
> > >
> > > 8. Negrea, J., Haghifam, M., Dziugaite, G.K., Khisti, A. and Roy, D.M., 2019. Information-theoretic generalization bounds for SGLD via data-dependent estimates. arXiv preprint arXiv:1911.02151.
> > >
> > > 9. Li, J., Luo, X. and Qiao, M., 2019. On generalization error bounds of noisy gradient methods for non-convex learning. arXiv preprint arXiv:1902.00621.

---

> > > > ### Comment · Reviewer_kujL · 2021-09-03
> > > > **score raised**
> > > >
> > > > Thanks for the further clarifications. My concern on the significance of Theorem 1 is about the result itself and its practicality, not about its difference with Theorem 2. But after the discussion with other reviewers and the area chair, I've raised the score.

---

### Official Review · Reviewer_42Mp · 2021-07-21

**Rating:** 7
**Confidence:** 4

**Summary:**

 The authors study the generalization properties of stochastic gradient Langevin dynamics (SGLD),  using
results about Gaussian channels from info theory.

The paper derives a generalization bound that involves the variance of the mini batch gradients in various steps. The variance terms are multiplied by a decaying factor (Thm 1) when one is interested in generalization bound for the last iterate, and the factor is (not surprisingly) absent when the function under consideration depends on all the iterates (Thm 2). The results are established by building on the recent results relating generalization bounds for sub-Gaussian loss functions in terms of the mutual information between the parameter iterates and samples used at various iterations (Bu et al. 2020 result), and upper bounding the mutual information in terms of the variance of gradient of the loss function, and the SGLD step size eta, and temperature beta (Lemma 5). The results scale as 1/n with sample size n---which is better than some of the previous results that try to remove the Lip constant dependence.

The authors present some numerical experiments that demonstrate that their generalization bounds correlate well with generalization gaps observed with small versions of MNIST and CIFAR 10 datasets (5k training examples): (a) with the amount of corrupted labels---increasing corruption increases the generalization error, (b) the width of the network---increasing width decreases the generalization. Their bounds (Table 1) correlate with generalization bounds better compared with Negrea et al. 2019 results, when one considers various settings across learning rate, width, and depth hyper-parameters.

The paper is very well written and fairly easy to follow---a feature missing in several submissions in recent times. It summarizes some useful results in a reader-friendly way.

**Limitations And Societal Impact:**

The authors have not included a discussion on this.

**Main Review:**


I have four major concerns with the current version of the work:

1. The authors say that prior works often characterize the generalization bounds in terms of the Lipschitz constants, and argue that '''such bounds can be potentially loose since the Lipschitz constants may be large and fails to explain some empirical observations (e.g., a network trained using true labels generalizes better than a network trained using corrupted labels as shown in Figure 1)'''--- I find this argument incomplete, especially because prior work by Foster et al. https://arxiv.org/pdf/1706.08498.pdf  (that this work does not cite) try to study phenomenon identical to Figure 1 and they show that Lipschitz based bounds can explain this phenomenon to a certain degree; the constants are large when there are fake labels. In fact, one of the key pieces of numerical evidence for the usefulness of the results in this draft is the same as the evidence in this earlier work---the generalization bounds ``correlate" with the generalization gaps observed in datasets with correct and fake labels. I would like to see a discussion to this end.


2. The authors state '''In contrast, we do not need any assumptions on the learning rate.''' This aspect while on the face seems true, it is unclear to me how the variance bounds in Thm 1 and 2 evolve over time for arbitrary step sizes. Do we not need some condition on eta for the variances to remain stable over time? For large eta, I can see that the q will degenerate to 1, and that we have a factor of sqrt(eta) in Thm 1, but the variance terms will begin to blow up as we increase eta. This leads to a slightly more important question: Some discussion is warranted on what condition on eta and beta is sufficient to ensure that the bounds in Thm 1 and Thm 2 are not vacuous, or increasing with time.

3. While it is heartening to see the bounds having some nice correlations, none of the experiments (I checked the appendix) involved a large sample dataset (the usual 50k for MNIST and CIFAR 10)---is there a reason for that? Also, the authors should add details about the variance of gradients were estimated?

4. The authors discuss related work earlier in the work, but ignore it completely after stating their bounds more formally---a revisit to the key results from prior works and putting their bounds in Thm 1 and Thm 2 would allow me to better appreciate their results in context. Also, see the minor question below.

Minor question (optional)

1. Are there settings under which Lemma 5 is tight? For instance, does the bound in Thm 1 provide a strict improvement over Lemma 3 https://arxiv.org/pdf/1901.04609.pdf for large T?

**Time Spent Reviewing:**

5

---

> ### Author Response · Authors · 2021-08-09
> **Response to the comments and concerns from Reviewer 42Mp**
>
> We thank the reviewer for providing thoughtful reviews of the paper and giving constructive feedback!
>
> $\textbf{[Main review]}$
>
> 1. This is a great question! To clarify, the Lipschitz constant we referred to is the Lipschitz constant of the loss function $\ell(w,z)$ w.r.t. the parameter $w$ (i.e., the neural network weight matrices). This quantity is different from the Lipschitz constant of the function $f_w(x)$ corresponding to a network w.r.t. the input variable $x$. For the sake of illustration, we denote the former Lipschitz constant by $Lip_w(\ell)$ and the latter one by $Lip_x(f)$. As the reviewer pointed out, $Lip_x(f)$ has been used in the literature, including [Bartlett, Foster, Telgarsky, 2017], for deriving generalization bounds and, to some degree, can capture generalization phenomena, such as label corruption. This is because a network trained from corrupted data has different weight matrices compared with the one trained from true data, leading to different values of $Lip_x(f)$. On the other hand, $Lip_w(\ell)$ is what previous literature used for bounding SGLD generalization error [see e.g., Theorem 1 in Mou et al., 2018; Pensia et al., 2018; Bu et al., 2020]. These bounds fail to explain label corruption because the Lipschitz constant $Lip_w(\ell) = \sup_{w,z} \|\nabla_w \ell(w,z)\|$ takes a supremum over all possible weight matrices $w$ and data point $z$. In other words, $Lip_w(\ell)$ only relies on the architecture of the network instead of the weight matrices or data distribution. Hence, these SGLD bounds give the same value for a network trained from corrupted data and a network trained from true data. Finally, $Lip_w(\ell)$ is often much larger than $Lip_x(f)$ because the weight matrices are in a higher dimension than the input variable. We will add the above clarification to our paper and include the reference suggested by the reviewer, along with other references which derive generalization bounds using a Lipschitz constant.
>
> 2. We include the sentence "In contrast, we do not need any assumptions on the learning rate."  in the context of a comparison with the bound in Li et al., 2019. Specifically, our generalization bound always holds regardless of the learning rate schedule while the bound in Li et al., 2019 is only applicable when the learning rate satisfies a particular assumption (see the second condition in their Theorem 11). Regarding the variance term, we observe empirically that it tends to be stable over time when the learning rate $\eta_t$ is chosen such that the SGLD algorithm converges. The intuition is that the variance of gradients reflects a certain kind of sharpness of the loss landscape. As the algorithm converges to a minimum, the loss landscape becomes flatter. You are correct in terms of the observation that large $\eta_t$ will lead to a trivial decay factor $q$. However, from a practical perspective, the learning rate is often bounded (or gradually decreases over time). Finally, a sufficient condition that can ensure that our bounds are stable is that $\eta_t \beta_t$ can be upper bounded, which is often satisfied in practice (e.g., we set $\eta_t \beta_t = 10^6/2$ in our experiments).
>
>
> 3. We chose 5k samples to reduce computational time but there is no problem applying our bound to a large sample dataset. For your second concern regarding how the variances were estimated, we will add the following experimental details to our revised paper.
>
> >We repeat our experiment 4 times and record the batch gradient at each iteration. This batch gradient is the one used for updating the parameters in the SGLD algorithm so it does not require any additional computations. Then we estimate the variance of gradients in Theorem 2 by using the population variance of the recorded batch gradients. Finally, we repeat the above procedure 4 times for computing the standard deviation, leading to the shaded areas in Figure 1.
>
> 4. We thank the reviewer for pointing out this issue and will include the following discussion after Corollary 1.
>
> >The order of our generalization bound in Corollary 1 is
> $$\min\left(\frac{1}{n} \sum_{t=1}^T \sqrt{\beta \eta_t}, \sqrt{\frac{\beta}{bn} \sum_{t=1}^T \eta_t} \right).$$ Our bound is distribution-dependent through the variance of gradients in contrast with [Pensia et al., 2018; Bu et al., 2020; stability bound in Mou et al., 2018]. It is also tighter than [PAC-Bayesian bound in Mou et al., 2018] whose order is $\sqrt{\frac{\beta}{n} \sum_{t=1}^T \eta_t}$. Our bound is applicable regardless of the choice of learning rate while the bound in [Li et al., 2019] requires the scale of the learning rate to be upper bounded by the reciprocal of the Lipschitz constant which, as already pointed out by [Negrea et al., 2019], limits its applicability to typical learning problems since the Lipschitz constant can be prohibitively large. Our Corollary 1 has the same order with [Negrea et al., 2019] but we incorporate an additional decay factor under additional assumptions and numerical experiments suggest that our bound is more favourably correlated with the true generalization gap. Finally, we remark that the purpose of introducing Corollary 1 is to derive a generalization bound that has a similar form with existing bounds so that we can compare. However, our main result (Theorem 2) is often tighter than Corollary 1, since this corollary is obtained by combining Theorem 2 with Minkowski inequality and Jensen's inequality.
>
>
>
>
>
>
> $\textbf{[Minor question]}$
>
> 1. Yes, we provide the following example to illustrate the sharpness of our Lemma 5. Compared with Lemma 3 and Proposition 3 in [Bu et al., 2020], the main improvement is to replace the Lipschitz constant by the variance of gradients. As we explained in the response to your first concern, this improvement not only enables the bound to be distribution-dependent but significantly tightens the bound (note that the Lipschitz constant used in [Bu et al., 2020] is a uniform upper bound of our variance of gradients).
>
> >[Sharpness of Lemma 5] Below we prove that the mutual information on the left-hand side can be decomposed as the variance of gradients and an additional term. The variance term matches the upper bound in Lemma 5 and the additional term converges to zero as the magnitude of the Gaussian noise is increasing. This demonstrates the tightness of the upper bound in Lemma 5. Let us focus on the first iteration (i.e., $t=1$). Assume that the initial point $W_0 = 0$; the first iteration only uses the data point $Z_1$ (i.e., $b=1$); the parameter domain $\mathcal{W} = R^d$. Furthermore, we assume that the gradient is bounded and has a zero expectation. Now we can use the golden formula (see Line 528 in Appendix) which yields
> $$I(W_1; Z_1) = \frac{\beta_t \eta_t}{4} \mathsf{Var}(\nabla_w \hat{\ell}(W_0, Z)) - D_{\text{KL}}\left(P_{-\sqrt{\frac{\eta_t \beta_t}{2}} \nabla_w \hat{\ell}(W_0, Z)+N} \| P_N\right).$$
> The first term on the right-hand side is exactly the upper bound in Lemma 5 (up to a $1/4$ constant difference). The second term converges to zero as $\beta_t \to 0$. To summarize, this example suggests that our upper bound in Lemma 5 is sharp when the magnitude of the additive noise is large enough.
>
> We hope that our response can answer all of your comments and would like to thank the reviewer again for providing constructive feedback!

---

> > ### Comment · Reviewer_42Mp · 2021-08-25
> > **Response to the authors' response**
> >
> > I am happy with the authors' response to all my major questions, and I thank the authors for providing the necessary arguments in detail. Assuming the suggested changes will be made in the final submission, I have increased my rating for this work.

---

### Decision · Program_Chairs · 2021-09-27

**Decision:**

Accept (Poster)

**Comment:**

The paper provides new generalization guarantees for SGLD, which incorporate the gradient variance, giving a more adaptive flavor. The paper also considers differentially private stochastic optimization as an application of their results.

We had quite a bit of discussion on this paper and we thank the authors for providing the reviewers with additional details, which helped to clear up some confusion. After deliberation, we agreed that the paper should be accepted.

Please do carefully go over the reviews and discussion and incorporate any suggestions into the final manuscript.